# MiniMax Learning of Interpretable Factored Stochastic Policies from Conjoint Data, with Uncertainty Quantification

**Connor T. Jerzak** [1]   **Priyanshi Chandra** [2]   **Rishi Hazra** [3]

## Abstract

We study offline policy optimization over exponentially large factorial action spaces from randomized preference data, showing how conjoint experiments can estimate interpretable stochastic policies with asymptotically valid uncertainty under regularity conditions. Conjoint analyses typically report Average Marginal Component Effects (AMCEs) by averaging over opponent attributes and thus ignore strategic interdependence. We instead learn *stochastic interventions*—product-of-Categorical policies over factor levels—that (i) optimize expected outcomes in an average-case setting and (ii) extend to a two-player *minimax* (adversarial) setting that realistically captures simultaneous strategic candidate selection. Methodologically, we derive a closed-form optimizer for a tractable two-way interaction regime with $L_2$ variance regularization, and provide a general gradient-based procedure for richer model classes. Uncertainty from the outcome model propagates asymptotically to both the optimal policy and its value via a Delta method approximation. We further model institutional details (e.g., primaries) inside the minimax objective and introduce a data-driven measure of strategic divergence between parties. On synthetic data, we empirically characterize finite-sample error and coverage as dimensionality and $n$ vary. On a U.S. presidential conjoint, adversarially learned policies produce restricted-equilibrium vote shares that align with historical election ranges in our data, in stark contrast to non-adversarial (averaging) optimizers.

[1]Department of Government, University of Texas at Austin, Austin, Texas, USA [2]Faculty of Informatics, Università della Svizzera Italiana, Lugano, Switzerland [3]Departments of Statistics and Economics, Harvard College, Cambridge, Massachusetts, USA. Correspondence to: Connor T. Jerzak <connor.jerzak@austin.utexas.edu>.

*Proceedings of the 43rd International Conference on Machine Learning*, Seoul, South Korea. PMLR 306, 2026. Copyright 2026 by the author(s).

## 1. Introduction

Over the past decade, conjoint analysis, which is an application of high-dimensional factorial design, has become the most popular survey experiment methodology to study multidimensional preferences. Widely deployed across disciplines, conjoint analysis is used to evaluate complex, multi-attribute subjects ranging from consumer products and job applicants to public policies and political candidate profiles (e.g., Franchino & Zucchini, 2015; Ono & Burden, 2019; Christensen et al., 2021; Kirkland & Coppock, 2018).

In such experiments, respondents are asked, for example, to choose between two hypothetical political candidates whose features (e.g., gender, race, age, education, partisanship, and policy positions) are randomly selected. This design often employs a forced-choice format, where respondents must select one of the two candidates without an option to abstain or express no preference (Abramson et al., 2023). Researchers then proceed by estimating the average causal effect of each feature while marginalizing the remaining features over a particular distribution of choice. This popular quantity of interest is termed the Average Marginal Component Effect (AMCE) (Hainmueller et al., 2014).

AMCEs average a feature's effect over a chosen distribution for the other features, which means the answer depends on that averaging choice (De la Cuesta et al., 2022). In practice, researchers often use a uniform distribution, but real candidate pools are not uniform, and candidates do not choose their profiles in strategic isolation from opponents.

We therefore replace effect estimation with *policy learning*. Instead of asking for the marginal effect of a single attribute, we learn a *factored stochastic policy* over full profiles: a mixed distribution that assigns independent categorical probabilities to each attribute level and thus remains interpretable (i.e., one can read off how much weight the policy puts on, say, "economy" versus "immigration" as policy priority). In the non-adversarial, average-case setting, this policy is chosen to maximize expected win probability against a fixed reference distribution for the opponent. We control variance and preserve interpretability by shrinking the learned policy toward the experimental assignment.

To capture strategic interaction, we extend to an adversarial

setting in which both sides simultaneously choose their own mixed profile distributions. The objective is minimax—each side selects a profile distribution that is best against the other's—and institutional details are built in. Specifically, we model two stages: primary elections within each party (which induce a distribution over nominees) followed by the general election. The resulting restricted-equilibrium policies reflect how strategic opponents and institutions jointly shape feasible profile choices.

Under a linear outcome model with two-way interactions, the average-case optimizer with squared-distance regularization has a closed-form solution; uncertainty for both the optimal policy and its value is obtained by propagating forward the outcome-model uncertainty using implicit differentiation. The same recipe admits richer models such as generalized linear models (GLMs) or Bayesian neural networks; the same machinery also applies when we move from average-case to adversarial, institution-aware games.

The machine-learning problem here is therefore offline policy optimization over a combinatorial discrete action space whose size grows exponentially with the number of attributes, not only the estimation of conjoint effects. The product-of-Categoricals restriction gives a tractable and attribute-readable policy class; the closed-form result characterizes a two-way regime where this optimization can be solved analytically, while the differentiable and minimax extensions cover richer GLM and neural objectives. The uncertainty layer connects the learned stochastic policy to trustworthy ML by propagating outcome-model uncertainty into both policy probabilities and values.

**In sum, our contributions are:** (1) A shift from AMCEs to a conjoint estimand that is a factored stochastic policy over profiles, learned to maximize expected performance, and interpretable at the attribute-level. (2) A closed-form average-case optimizer under two-way interactions with squared-distance variance control, together with uncertainty quantification for both the optimal policy and its value via the Delta method. (3) A general gradient-based procedure for richer outcome models (including regularized GLMs and neural models) with end-to-end differentiation for standard errors. (4) An adversarial, minimax extension that embeds institutional structure (primaries then general), yielding restricted-equilibrium mixed strategies and a data-driven measure of strategic divergence between parties. (5) Empirical evidence from simulations and a U.S. presidential conjoint showing that adversarially learned policies produce restricted-equilibrium vote shares aligned with historical ranges, plus an open-source mapping of historical candidate features to conjoint levels to facilitate replication.

## 2. Background: Conjoint as Policy Learning

**Conjoint Experiments.** Conjoint analysis is a factorial survey experiment in which respondents evaluate multi-attribute profiles. In a typical forced-choice design, each respondent sees two hypothetical profiles (e.g., political candidates with attributes like age, gender, policy positions) and must select one as preferred. The experimenter randomizes attribute levels independently across profiles, often with uniform probability—this randomization serves as the *logging policy* in the language of offline learning.

**Standard Estimands: AMCE and AMIE.** The dominant approach estimates the Average Marginal Component Effect (AMCE): the expected change in selection probability when a single attribute shifts from baseline to treatment level, averaging over all other attributes (Hainmueller et al., 2014). Extensions include the Average Marginal Interaction Effect (AMIE) for two-way interactions (Egami & Imai, 2019). These estimands describe *marginal* effects under a chosen marginalization distribution, answering "what is the effect of changing one component?"

**From Effect Estimation to Policy Optimization.** We take a different perspective: instead of estimating marginal effects, we learn a *policy*—a distribution over full profiles—that maximizes expected outcomes. This shift matters when attribute interactions are present. Consider a hiring conjoint where "quantitative skills" and "communication skills" each have positive main effects, but their interaction is negative (substitutes). An AMCE-based approach might select both attributes at high levels, whereas the optimal policy may diversify, placing probability mass on profiles emphasizing one skill or the other.

**Connection to Offline Policy Learning.** Our framework maps naturally to offline contextual bandits with combinatorial actions:

- **Profile** $\mathbf{t} \in \mathcal{T}$: combinatorial action vector (the full attribute configuration)
- **Conjoint randomization** $\mathrm{Pr}_{\mathbf{p}}$: logging policy
- **Choice outcome** $C$: reward (0/1 selection indicator)
- **Factored policy** $\mathrm{Pr}_{\boldsymbol{\pi}}$: interpretable product-of-Categoricals policy class
- **Average-case**: offline policy learning against a fixed opponent/environment
- **Adversarial**: zero-sum game where both players optimize simultaneously

Because profile-pair assignment is randomized, conjoint data sidestep behavior-policy confounding that arises in observational offline RL (Levine et al., 2020), while still requiring the outcome-model, positivity, and policy-class assumptions stated below. Counterfactual outcomes can then be evaluated via direct modeling or off-policy estimators (Dudik et al., 2011). Classical zero-sum results and

**Example 1.** Consider a simple conjoint with $D = 2$ factors and $L = 2$ levels each:
- *Experience*: {Outsider, Insider}
- *Ideology*: {Moderate, Hardline}

A **profile** $\mathbf{t} = $ (Outsider, Moderate) is one of $2^2 = 4$ possible attribute combinations.

A **factored stochastic policy** $\boldsymbol{\pi}$ assigns independent probabilities: e.g., $\pi_{\text{Exp}}(\text{Outsider}) = 0.7$ and $\pi_{\text{Ideo}}(\text{Moderate}) = 0.6$, so $\Pr_{\boldsymbol{\pi}}(\text{Outsider}, \text{Moderate}) = 0.7 \times 0.6 = 0.42$.

**Why not just maximize AMCEs?** Suppose "Outsider" has a positive main effect (+5%) and "Moderate" also has a positive effect (+3%), but their interaction is negative ($-6\%$)—Outsider-Moderates are seen as lacking credibility. An AMCE-based policy would select (Outsider, Moderate), ignoring the interaction. Our method accounts for interactions and may instead place probability mass on (Outsider, Hardline) or (Insider, Moderate), yielding higher expected outcomes.

modern first-order minimax methods usually optimize over full mixed strategies with known utilities (Littman, 1994; Nemirovski, 2004); here utilities are estimated from randomized conjoint data, uncertainty is carried forward, and the target is a restricted factored minimax policy. §B contains additional related-work context.

# 3. Methods: Learning Factored Stochastic Policies from Conjoint Data

We first define the forced-choice potential outcomes and the randomization assumptions that identify counterfactual choice probabilities. We then move from an average-case policy objective to factored stochastic policies, outcome modeling, and the closed-form two-way optimizer. The remaining subsections describe differentiable optimization with uncertainty propagation, the adversarial minimax extension, and strategic-divergence diagnostics.

**Setup & Identification.** Suppose that we have a simple random sample of $n$ respondents from a population. We consider a conjoint design of candidate choice with a total of $D$ factorial features per candidate. Each factorial feature $d \in \{1, 2, \ldots, D\}$ has $L_d \geq 2$ levels. The random variable representing an entire candidate profile presented to respondent $i$ in the design is labeled $\mathbf{T}_i$. The support of $\mathbf{T}_i$, denoted $\mathcal{T}$, is the space of all possible treatment assignments and will vary based on the experimental design. For example, if each feature has $L$ levels, i.e., $L_1 = \cdots = L_D = L$, we have $\mathbf{t} \in \mathcal{T} = \{1, 2, ..., L\}^D$, where $\mathbf{t}$ is a specific realization of $\mathbf{T}_i$.

Usually, each respondent $i$ faces a choice between two candidate profiles, $\mathbf{T}_i^a$ and $\mathbf{T}_i^b$. Let $m_i(\mathbf{t})$ denote respondent $i$'s systematic utility for profile $\mathbf{t}$. Realized arm-specific utilities in a forced-choice task are

$$U_i^a(\mathbf{t}) = m_i(\mathbf{t}) + \varepsilon_i^a, \qquad U_i^b(\mathbf{t}') = m_i(\mathbf{t}') + \varepsilon_i^b,$$

where the shocks are continuous and exchangeable across arms, conditional on profiles and respondent characteristics. The pairwise potential choice outcome is

$$C_i(\mathbf{t}, \mathbf{t}') = \mathbb{I}\{U_i^a(\mathbf{t}) > U_i^b(\mathbf{t}')\}.$$

The observed outcome is $C_i = C_i(\mathbf{T}_i^a, \mathbf{T}_i^b)$. This represents the standard paired-profile conjoint in which respondents must select exactly one of two profiles as preferable (Hainmueller et al., 2014). We use $C_i$ for forced-choice outcomes; $Y_i$ denotes generic policy-learning outcomes outside the forced-choice notation. Ties have probability 0 (due to the shocks). If $\mathbf{t} = \mathbf{t}'$, exchangeability implies $\Pr\{C_i(\mathbf{t}, \mathbf{t}) = 1\} = 1/2$. More generally, let $\Delta \varepsilon_i := \varepsilon_i^b - \varepsilon_i^a$; let $F_\Delta$ denote its conditional CDF. Then,

$$\Pr\{C_i(\mathbf{t}, \mathbf{t}') = 1 \mid \mathbf{t}, \mathbf{t}'\} = F_\Delta\{m_i(\mathbf{t}) - m_i(\mathbf{t}')\}.$$

In the standard logit special case, $F_\Delta = \sigma$; for example, this arises when the arm-specific shocks are i.i.d. type-I extreme-value with common scale, so that the shock difference is logistic. With unit scale,

$$\Pr\{C_i(\mathbf{t}, \mathbf{t}') = 1 \mid \mathbf{t}, \mathbf{t}'\} = \sigma\{m_i(\mathbf{t}) - m_i(\mathbf{t}')\},$$
$$\sigma(x) = \{1 + \exp(-x)\}^{-1}.$$

Often, each treatment combination is equally likely to be realized, in which case $\Pr(\mathbf{T}_i = \mathbf{t}) = |\mathcal{T}|^{-1}$ for all treatment combinations, $\mathbf{t}$. When factor levels have possibly different assignment probabilities, some treatment combinations will be more likely than others. Often, each factor is assigned using draws from independent Categorical distributions so we can write the probability of treatment combination $\mathbf{t}$:

$$\Pr(\mathbf{T}_i = \mathbf{t}) = \prod_{d=1}^{D} \prod_{l=1}^{L_d} p_{dl}^{\mathbb{I}\{t_d = l\}},$$

where $p_{dl}$ is the Categorical probability for factor $d$ taking on level $l$ and $\mathbb{I}\{t_d = l\}$ is the indicator function that is 1 when $t_d$ takes on value $l$ and 0 otherwise. We let $\mathbf{p}$ define the vector of Categorical probabilities defining the data-generating distribution.

For simplicity, we make standard assumptions. We assume that there is no interference between units and that the profile-pair assignment is randomized, i.e., $\{C_i(\mathbf{t}, \mathbf{t}') : \mathbf{t}, \mathbf{t}' \in \mathcal{T}\} \perp\!\!\!\perp \{\mathbf{T}_i^a, \mathbf{T}_i^b\}$, with positive probability on all admissible profile pairs in the experimental design.

**Average-Case Policy Objective.** One approach in the policy learning literature is to identify the following optimal treatment combination, $\mathbf{t}^* = \arg\max_{\mathbf{t} \in \mathcal{T}} \mathbb{E}[Y_i(\mathbf{t})]$, where $\mathbf{t}^*$ is the treatment combination that maximizes the average value of some generic outcome, $Y_i$. In the forced-choice conjoint case, this quantity would amount to $\mathbf{t}^{a^*} = \arg\max_{\mathbf{t}^a \in \mathcal{T}} \mathbb{E}[C_i(\mathbf{t}^a, \mathbf{T}_i^b)]$, so investigators find

the vote-share-maximizing candidate profile $\mathbf{t}^{a^*}$, averaging over opposing candidate $b$ features (as in AMCE analysis). This approach has two limitations. First, high-dimensional treatments in conjoint analysis prevent reliably estimating $\mathbf{t}^*$ nonparametrically, as $|\mathcal{T}|$ far exceeds the sample size. Second, when multiple equally optimal profiles exist, learning several is more informative than a single one.

To address this challenge, we propose finding an optimal stochastic intervention: we consider a parametric distribution of profiles $\Pr_{\boldsymbol{\pi}}(\cdot)$ that maximizes the average outcome. By considering a parametric model, we are able to effectively summarize a set of profiles that perform well. Formally, we seek the optimal stochastic intervention,

$$Q(\boldsymbol{\pi}^*) = \max_{\boldsymbol{\pi}} Q(\boldsymbol{\pi}),$$
$$Q(\boldsymbol{\pi}) = \sum_{\mathbf{t} \in \mathcal{T}} \mathbb{E}[Y_i(\mathbf{t})] \Pr_{\boldsymbol{\pi}}(\mathbf{T}_i = \mathbf{t}). \quad (1)$$

where $\boldsymbol{\pi}$ parameterizes the distribution of profiles. In the forced-choice conjoint case, this quantity can be written in the average case as agent $A$ optimizing their strategy, averaging over $B$'s fixed strategy:

$$Q(\boldsymbol{\pi}^a) = \sum_{\mathbf{t}^a, \mathbf{t}^b \in \mathcal{T}} \mathbb{E}\Big[C_i(\mathbf{t}^a, \mathbf{t}^b)\Big] \Pr_{\boldsymbol{\pi}^a}(\mathbf{T}_i^a = \mathbf{t}^a)\Pr_{\mathbf{p}}(\mathbf{T}_i^b = \mathbf{t}^b).$$

$$\boldsymbol{\pi}^{a*} \in \arg\max_{\boldsymbol{\pi}^a} Q(\boldsymbol{\pi}^a).$$

Here, $\boldsymbol{\pi}^{a^*}$ characterizes the highest possible vote share for a given counterfactual strategy of assigning the candidate characteristics of $a$, while features of $b$ are assigned according to a static averaging distribution (e.g., uniform).

**Restricted Factored Policies.** Building on Equation 1, we preserve interpretability by restricting the counterfactual profile distribution for candidate $a$ to the same product-of-Categoricals used by the conjoint randomization $\Pr_{\mathbf{p}}$. Concretely, $\Pr_{\boldsymbol{\pi}^a}(\mathbf{T}_i^a = \mathbf{t}^a)$ has the identical factorized form as $\Pr_{\mathbf{p}}(\mathbf{T}_i = \mathbf{t})$, but with per-attribute probabilities $\boldsymbol{\pi}^a$ replacing $\mathbf{p}$. This choice yields an attribute-readable policy and keeps off-policy evaluation tractable. The deterministic "best profile" appears as a degenerate special case, $\Pr_{\boldsymbol{\pi}^*}(\mathbf{T}_i = \mathbf{t}) = \mathbb{I}(\mathbf{t} = \mathbf{t}^*)$, but in high-dimensional conjoints that target is not reliably estimable and is statistically brittle; we therefore optimize over *stochastic* policies that summarize families of high-performing profiles.

To allow meaningful deviations from the design while controlling variance, we impose an $L_2$ (or KL) trust-region around the logging distribution:

$$\max_{\boldsymbol{\pi}^a} Q(\boldsymbol{\pi}^a) - \lambda_n \|\boldsymbol{\pi}^a - \mathbf{p}\|_2^2,$$

equivalently constraining $\|\boldsymbol{\pi}^a - \mathbf{p}\|_2 \leq \epsilon_n$. This regularization is motivated by the increase in off-policy variance as $\boldsymbol{\pi}^a$ departs from $\mathbf{p}$. The restriction–regularization pair—matching the conjoint's factorized assignment and shrinking

toward $\mathbf{p}$—is important because it (i) preserves interpretability, (ii) stabilizes estimation, and (iii) yields a closed-form average-case optimizer under two-way interactions (Proposition 3.1), while still admitting general gradient-based solutions for richer (e.g., neural) outcome models.

**Restricted Policies as Variational Approximations.** When the variance-control regularizer is KL, the full-simplex optimal policy has the closed-form Gibbs/log-linear form $\sigma^\star(\mathbf{t}) \propto p(\mathbf{t}) \exp\{u(\mathbf{t})/\lambda\}$ for an appropriate utility $u(\mathbf{t})$. Restricting to product-of-Categorical policies is exactly the classical mean-field (product) variational approximation to this Gibbs distribution (the theoretical, unrestricted "perfect" strategy). In §A.4, we give an explicit bound for KL-regularized objectives. We also give an exploitability decomposition that finds the maximum unilateral payoff improvement available against a learned opponent.

### 3.1. Outcome Models

**Bernoulli GLM.** Let $C_i = C_i(\mathbf{T}_i^a, \mathbf{T}_i^b)$ denote the forced choice. We model $C_i \mid (\mathbf{T}_i^a, \mathbf{T}_i^b) \sim \text{Bernoulli}(\sigma(\eta_i))$, where $\sigma(x) = \{1 + \exp(-x)\}^{-1}$ is the logistic link. Write $I_i^c(dl) = \mathbb{I}\{T_{id}^c = l\}$ and $I_i^c(dl, d'l') = \mathbb{I}\{T_{id}^c = l, T_{id'}^c = l'\}$. Then

$$\eta_i = \sum_{d=1}^{D} \sum_{l=1}^{L_d} \beta_{dl} \left( I_i^a(dl) - I_i^b(dl) \right)$$
$$+ \sum_{d<d'} \sum_{l=1}^{L_d} \sum_{l'=1}^{L_{d'}} \gamma_{dl,d'l'} \left( I_i^a(dl, d'l') - I_i^b(dl, d'l') \right). \quad (2)$$

where $\beta_{dl}$ denotes the main effect of factor $d$ with level $l$, $\gamma_{d'l',d''l''}$ denotes the interaction effect of treatment $d'l'$ and $d''l''$. We impose sum-to-0 constraints on $\{\beta_{dl}\}_l$ and on each $\{\gamma_{dl,d'l'}\}_{l,l'}$ for identifiability (Egami & Imai, 2019). The intercept is constrained to 0 here to enforce the forced-choice antisymmetry condition. $\eta_i(\mathbf{t}, \mathbf{u}) = -\eta_i(\mathbf{u}, \mathbf{t})$, which implies $\Pr\{C_i(\mathbf{t}, \mathbf{t}) = 1\} = 1/2$ for identical profiles. Parameters $(\boldsymbol{\beta}, \boldsymbol{\gamma})$ are estimated via GLM, with optional sparsity (lasso) for selection followed by an unpenalized refit when inference is required. An intuition here is that the difference between utilities under candidates $a$ and $b$ defines the choice between $a$ and $b$. When the linear-in-indicators GLM is too restrictive, a natural alternative modeling choice would be a Bayesian neural network; for details, see §C.4.

**Scope of Parametric Assumptions.** The linear probability approximation below is used in the closed-form two-way optimizer; it is not a framework-wide requirement. The same policy objective can be optimized directly under Bernoulli GLMs and Bayesian neural outcome models by gradient methods, with uncertainty propagated through the fitted outcome model. Linear-in-indicator specifications

are standard starting points for randomized forced-choice conjoint analysis (Hainmueller et al., 2014; Egami & Imai, 2019); later robustness results in Table 2 quantify sensitivity to response surface nonlinearity.

### 3.2. Closed-Form Average-Case Optimizer

For the Bernoulli GLM, the average outcome $\mathbb{E}_{\boldsymbol{\pi}^a, \boldsymbol{\pi}^b}[\sigma\{\eta(\mathbf{T}^a, \mathbf{T}^b)\}]$ is optimized numerically as described below. The closed-form result in this subsection applies to the corresponding linear-probability approximation, where the conditional choice probability is linear in the profile indicators. Then, the *Stochastic Intervention Under Forced Choice Conjoint* can be written:

$$Q(\boldsymbol{\pi}^a, \boldsymbol{\pi}^b) = \mathbb{E}_{\mathbf{T}^a \sim \boldsymbol{\pi}^a, \, \mathbf{T}^b \sim \boldsymbol{\pi}^b} \left[ \sigma\Big(\eta(\mathbf{T}^a, \mathbf{T}^b)\Big) \right],$$

$$\eta(\mathbf{T}^a, \mathbf{T}^b) = \sum_{d,l} \beta_{dl} \left( I_{dl}^a - I_{dl}^b \right)$$
$$+ \sum_{d<d'} \sum_{l,l'} \gamma_{dl,d'l'} \left( I_{dl,d'l'}^a - I_{dl,d'l'}^b \right),$$

where $I_{dl}^c$ and $I_{dl,d'l'}^c$ are the corresponding main-effect and two-way indicators for a draw from policy $c$. Under a linear probability approximation, this becomes:

$$Q(\boldsymbol{\pi}^a, \boldsymbol{\pi}^b) = \mathbb{E}_{\boldsymbol{\pi}^a, \boldsymbol{\pi}^b} \left[ \Pr\{C_i(\mathbf{T}_i^a, \mathbf{T}_i^b) = 1\} \right]$$

$$= \frac{1}{2} + \sum_{d=1}^{D} \sum_{l=1}^{L_d} \beta_{dl} \left( \pi_{dl}^a - \pi_{dl}^b \right)$$

$$+ \sum_{d<d'} \sum_{l=1}^{L_d} \sum_{l'=1}^{L_{d'}} \gamma_{dl,d'l'} \, \pi_{dl}^a \pi_{d'l'}^a$$

$$- \sum_{d<d'} \sum_{l=1}^{L_d} \sum_{l'=1}^{L_{d'}} \gamma_{dl,d'l'} \, \pi_{dl}^b \pi_{d'l'}^b.$$

The constant $1/2$ is fixed by the forced-choice symmetry condition and does not affect the optimizer, so it is omitted from the stationary first-order system below. Motivated by the opponent candidate marginalization in AMCE analysis, we first consider the optimal average-case stochastic intervention where $a$ optimizes against a uniform distribution over candidate features. In this case, $\boldsymbol{\pi}^{a^*}$ can be derived in closed form, assuming the features of the opposing candidate, $b$, are assigned according to a fixed distribution such as $\mathbf{p}$. We call this kind of analysis the *Average Case Optimal Stochastic Intervention for Forced-Choice Conjoints* in that the opponent, $b$, is static.

Before applying the closed-form result, we re-express the linear-probability approximation in baseline coordinates. Let level $L_d$ be the baseline for factor $d$, and write $x_{dl} = \mathbb{I}\{t_d = l\}$ only for $l < L_d$. If the outcome model is estimated with sum-to-zero main effects $\beta_{dl}$ and pairwise interactions $\gamma_{dl,d'l'}$, define the baseline-coded coefficients

$(\bar\beta, \bar\gamma)$ by

$$\bar\gamma_{dl,d'l'} = \gamma_{dl,d'l'} - \gamma_{dL_d,d'l'} - \gamma_{dl,d'L_{d'}}$$
$$+ \gamma_{dL_d,d'L_{d'}}, \qquad l < L_d, \; l' < L_{d'}.$$

and

$$\bar\beta_{dl} = \beta_{dl} - \beta_{dL_d}$$
$$+ \sum_{d'>d} \left( \gamma_{dl,d'L_{d'}} - \gamma_{dL_d,d'L_{d'}} \right)$$
$$+ \sum_{d'<d} \left( \gamma_{d'L_{d'},dl} - \gamma_{d'L_{d'},dL_d} \right), \qquad l < L_d.$$

The closed-form result below is stated in terms of these baseline-coded coefficients.

**Proposition 3.1.** *Under the baseline-coded linear-probability approximation with two-way interactions, let $(\bar\beta, \bar\gamma)$ denote either the coefficients obtained from a direct baseline-coded fit or the baseline-coordinate transformation of the sum-to-zero coefficients above. Taking level $L_d$ as the baseline so that $\pi_{dL_d} = 1 - \sum_{l<L_d} \pi_{dl}$, the equality-constrained stationary point of the average-case $L_2$-regularized objective satisfies, for each $d$ and $l \in \{1, \ldots, L_d - 1\}$,*

$$\boldsymbol{\pi}^{a^*} = \mathbf{C}^{-1}\mathbf{B},$$
$$B_{r(dl),1} = -\bar\beta_{dl} - 4\lambda_n p_{dl} - 2\lambda_n \sum_{l' \neq l, \, l' < L_d} p_{dl'},$$
$$C_{r(dl),r(dl)} = -4\lambda_n,$$
$$C_{r(dl),r(dl')} = -2\lambda_n,$$
$$C_{r(dl),r(d'l'')} = \bar\gamma_{dl,d'l''}.$$

*For $d \neq d'$, set $C_{r(dl),r(d'l')} = C_{r(d'l'),r(dl)} = \bar\gamma_{dl,d'l'}$. Here $r(dl)$ denotes an indexing function returning the position associated with its factor $d$ and level $l$ into the rows of $B$ and rows or columns of $C$. If $\lambda_n$ is large enough that the (symmetric) Hessian matrix is negative definite (equivalently, $\mathbf{C}$ as defined above is symmetric negative definite) and the solution lies in the simplex, then this stationary point is the unique global maximizer. For proof, see §A.2.*

If the linear-system solution violates non-negativity or lies on the simplex boundary, the displayed stationary system should not be interpreted as the constrained optimizer. In that case, one must solve the corresponding quadratic program with simplex constraints, or use the logit-parameterized gradient method described below.

Here, the optimal stochastic intervention, $\Pr_{\boldsymbol{\pi}^{a^*}}$, is a deterministic function of the outcome model parameters. The parameters defining the outcome model, $\boldsymbol{\beta}$ and $\boldsymbol{\gamma}$, are not known *a priori*, but can be estimated via GLM, with uncertainties calculated using asymptotic SEs from the unpenalized model. If a post-selection refit is used, these standard errors condition on the selected specification unless a valid post-selection or sample-splitting procedure is used.

Intuitively, the analysis done here allows researchers to investigate the implications of models for candidate choice fit on the data. Instead of examining marginals via AMCE, they can examine joint effects by looking at the optimal behavior implied under their choice of model. Estimates of the optimal distribution over candidates are generated using uncertain model parameters; the Delta method enables asymptotic propagation of that uncertainty.

As the values, $\boldsymbol{\pi}^{a^*}$ defining $\mathrm{Pr}_{\boldsymbol{\pi}^{a^*}}$ are a deterministic function of modeling parameters, the variance-covariance matrix of $\{\widehat{Q}(\widehat{\boldsymbol{\pi}}^{a^*}), \widehat{\boldsymbol{\pi}}^{a^*}\}$ can be obtained via the Delta method:

$$\text{Var-Cov}(\{\widehat{Q}(\widehat{\boldsymbol{\pi}}^{a^*}), \widehat{\boldsymbol{\pi}}^{a^*}\}) = \mathbf{J}\,\widehat{\Sigma}\,\mathbf{J}',$$

where $\widehat{\Sigma}$ is the variance-covariance matrix from the modeling strategy for $C_i$ using regression parameters $\{\boldsymbol{\beta}, \boldsymbol{\gamma}\}$ and $\mathbf{J}$ is the Jacobian of partial derivatives (e.g., of $\widehat{Q}(\widehat{\boldsymbol{\pi}}^{a^*})$ and $\widehat{\boldsymbol{\pi}}^{a^*}$ w.r.t. the outcome model parameters): $\mathbf{J} = \nabla_{\{\hat{\boldsymbol{\beta}}, \hat{\boldsymbol{\gamma}}\}}\{\widehat{Q}(\widehat{\boldsymbol{\pi}}^{a^*}), \widehat{\boldsymbol{\pi}}^{a^*}\}$. Assume (i) the first-stage estimator is asymptotically normal under i.i.d. sampling and correct specification, (ii) the regularized objective has a unique interior maximizer and the map from outcome-model parameters to $\boldsymbol{\pi}^{a^*}$ is differentiable at the truth, and (iii) optimization error is $o_p(n^{-1/2})$. Under these regularity conditions,

$$\sqrt{n}\left(\{\widehat{Q}(\widehat{\boldsymbol{\pi}}^{a^*}), \widehat{\boldsymbol{\pi}}^{a^*}\} - \{Q(\boldsymbol{\pi}^{a^*}), \boldsymbol{\pi}^{a^*}\}\right) \rightarrow \mathcal{N}\left(\mathbf{0}, \mathbf{J}\Sigma\mathbf{J}'\right).$$

A more formal statement covering both the closed-form and differentiable-optimization cases, including interiority conditions, is given in §A.6.

### 3.3. General Differentiable Optimization & Uncertainty

There are limitations to the approach just described: the analytical solution in Proposition 3.1 does not guarantee the non-negativity of $\hat{\boldsymbol{\pi}}^{a^*}$ for small values of $\lambda_n$. Also, as soon as we generalize the outcome model to the GLM or $>$ 2-way interactions, we have no currently known analytical formula for the optimal solution.

To address these limitations, we can perform the stochastic intervention optimization for $\widehat{\boldsymbol{\pi}}^{a^*}$ using iterative methods instead of an analytical closed form. For example, to ensure that the entries in $\hat{\boldsymbol{\pi}}^{a^*}$ lie on the simplex, we can re-parameterize the objective function using $\alpha_{dl}$'s, which inhabit an unconstrained space. Let $Z_d(\boldsymbol{\alpha}) = 1 + \sum_{l'=1}^{L_d-1} \exp(\alpha_{dl'})$ and $\mathrm{Pr}_{\boldsymbol{\pi}(\boldsymbol{a})}(T_d = l) = \pi_{dl}(\boldsymbol{a})$, where

$$\pi_{dl}(\boldsymbol{a}) = \begin{cases} \exp(\alpha_{dl})/Z_d(\boldsymbol{\alpha}) & \text{if } l < L_d, \\ 1/Z_d(\boldsymbol{\alpha}) & \text{if } l = L_d \text{ (baseline)}. \end{cases}$$

For the average-case one-player objective, standard compactness/boundedness and step-size conditions imply that limit points of projected gradient methods are stationary;

under additional strict-saddle/nondegeneracy assumptions, random initialization avoids strict saddles with probability one (Lee et al., 2016). We update the unconstrained parameters using the gradient information, $\nabla_{\boldsymbol{\alpha}}\{O(\boldsymbol{a})\}$. In particular, for $s = 0, \dots, S-1$,

$$\boldsymbol{\alpha}^{(s+1)} := \boldsymbol{\alpha}^{(s)} + \gamma^{(s)}\nabla_{\boldsymbol{a}}\{O(\boldsymbol{\alpha}^{(s)})\}.$$

For Delta-method inference, gradients are traced through all $S$ updates when computing the Jacobian $\mathbf{J}$. Approximate inference proceeds analogously to the closed-form case via a first-order Delta method. With a closed-form expression for $\hat{\boldsymbol{\pi}}^{a^*}$, it is evident how we could write an expression for the derivative of optimal as a function of the regression parameters using the closed-form Jacobian.

With an iterative computation needed to obtain $\hat{\boldsymbol{\pi}}^{a^*}$, we can consider the same quantity: although the closed-form derivatives of the iterative solution may be unknown, we can still evaluate these values using automatic differentiation—tracing the gradient information through the entire sequence of $S$ gradient ascent updates. Specifically, since the ascent procedure defines a deterministic mapping from $\hat{\boldsymbol{\beta}}, \hat{\boldsymbol{\gamma}}$ to $\widehat{\boldsymbol{\pi}}^{a^*}$ (and thence to $\widehat{Q}(\widehat{\boldsymbol{\pi}}^{a^*})$) for fixed $S$, reverse-mode differentiation can backpropagate through the full unrolled sequence of $S$ updates, yielding $\mathbf{J}$.

By contrast, we can avoid differentiating through a long optimization trace by applying *implicit differentiation* to the stationarity conditions at the converged solution. Let $F(\boldsymbol{\alpha}, \theta) = \nabla_{\boldsymbol{\alpha}} O(\boldsymbol{\alpha}; \theta)$ denote the first-order optimality mapping (in the adversarial case, stack the signed gradients for both players), where $\theta$ are the outcome-model parameters. If $F$ is continuously differentiable and the Jacobian $H = \nabla_{\boldsymbol{\alpha}} F(\hat{\boldsymbol{\alpha}}^*, \hat{\theta})$ is nonsingular (so the fixed point is locally isolated with an invertible Hessian/Jacobian), then the implicit function theorem gives $\partial\boldsymbol{\alpha}^*/\partial\theta = -H^{-1}\nabla_\theta F$. Consequently, the Delta-method Jacobian $\mathbf{J}$ for $(\boldsymbol{\pi}^*, Q(\boldsymbol{\pi}^*))$ can be obtained by solving a linear system in $H$ rather than backpropagating through all $S$ gradient steps (Lorraine et al., 2020). This "implicit" approach is typically much cheaper and more memory-efficient for long trainings and agrees with unrolling when the optimizer has converged to the same fixed point, but it can be unstable when optimization has not converged, $H$ is ill-conditioned/singular, or the solution lies near a boundary.

### 3.4. Adversarial Minimax Policies

Thus far, we have considered optimal stochastic interventions under the assumption that one party (or candidate) chooses its profile distribution to maximize expected vote share, while treating the distribution of the opposing candidate's profile as fixed. Although this framework is useful in settings without direct strategic interaction (e.g., analyzing hiring choices), it is less suitable when two agents strategi-

cally select their own profiles in direct competition. In many contexts, both the focal candidate and the opposing candidate are engaged in simultaneous strategic optimization.

To capture these adversarial dynamics, we introduce an *Adversarial Case Optimal Stochastic Intervention* framework that explicitly models two agents, which we label as $A$ and $B$, each attempting to maximize their expected probability of victory in a forced-choice setting. This is a two-player, simultaneous action zero-sum game. Let $m_i(\mathbf{T}_i^c)$ represent respondent $i$'s systematic utility for candidate $c \in \{A, B\}$, where $\mathbf{T}_i^c$ is the candidate's profile randomly drawn from some distribution. Under the same exchangeable shock formulation as above, the pairwise forced-choice potential outcome is:

$$C_i(\mathbf{t}, \mathbf{u}) = \mathbb{I}\{U_i^A(\mathbf{t}) > U_i^B(\mathbf{u})\}.$$

We define candidate profile distributions for $A$ and $B$ as $\boldsymbol{\pi}^A$ and $\boldsymbol{\pi}^B$, respectively. Each distribution assigns probabilities to the set of all possible profiles, $\mathcal{T}$. The choice of a stochastic (mixed) rather than deterministic profile stems from the combinatorics of potential profiles and impossibility of estimating a single optimal profile with finite samples.

We consider a zero-sum environment where one candidate's gain is the other's loss. Here, we characterize the optimal profile distributions through a min-max optimization problem. Letting $Q(\boldsymbol{\pi}^A, \boldsymbol{\pi}^B) = \mathbb{E}_{\boldsymbol{\pi}^A, \boldsymbol{\pi}^B}\left[C_i(\mathbf{T}_i^A, \mathbf{T}_i^B)\right]$ denote the expected probability that candidate $A$ wins against candidate $B$, the adversarial objective is:

$$\max_{\boldsymbol{\pi}^A} \min_{\boldsymbol{\pi}^B} Q(\boldsymbol{\pi}^A, \boldsymbol{\pi}^B). \tag{3}$$

In the unrestricted simplex, neither candidate can improve their expected performance by unilaterally changing their distribution. Such a pair $(\boldsymbol{\pi}^{A^*}, \boldsymbol{\pi}^{B^*})$ constitutes a Nash equilibrium for the adversarial environment, evoking classic results in game theory (Kreps, 1989). In other words, given $\boldsymbol{\pi}^{B^*}$, no deviation from $\boldsymbol{\pi}^{A^*}$ improves $A$'s performance, and vice versa. In our restricted factored class, we instead target stationary points of the restricted minimax objective.

**Institutional Constraints.** Without institutional asymmetries, the adversarial game just described can admit trivial or symmetric equilibria. Real strategic environments (such as elections), however, are structured by rules that determine how agents interact (e.g., *who* votes *when*), therefore shaping both strategies and equilibria. We model a general two-stage system—for example, party primaries followed by a general election—under potentially asymmetric institutions across competitors.

Let $A$ and $B$ index the two competing agents (e.g., political parties). Denote by $\mathcal{I}^A$ and $\mathcal{I}^B$ the first-stage participant sets for $A$ and $B$, respectively (in our electoral example,

the possibly overlapping primary electorates, where closed, semi-open, and open primaries are special cases), and by $\mathcal{E}$ the final-stage participant set (the general-election electorate). Institutions determine these sets and their sampling weights (e.g., turnout, inclusion of independents); we bundle these parameters as $\beth$. Each agent $c \in \{A, B\}$ chooses a factored, product-of-Categorical mixed profile distribution $\boldsymbol{\pi}^c$ (our policy); the rest of that agent's field is summarized by a counter-distribution $\boldsymbol{\pi}^{c'}$.

**Institutionalized Value.** Given nominees $\mathbf{t} \sim \bar{\boldsymbol{\pi}}^A$ and $\mathbf{u} \sim \bar{\boldsymbol{\pi}}^B$, the probability that $A$ wins the general election—averaging over the general electorate $\mathcal{E}$—is

$$V(\mathbf{t}, \mathbf{u}) := \Pr\{C_i(\mathbf{t}, \mathbf{u}) = 1 \mid i \in \mathcal{E}\}$$
$$= \mathbb{E}_{i \in \mathcal{E}}\left[G\{m_i(\mathbf{t}) - m_i(\mathbf{u})\}\right],$$

where $G = F_\Delta$ in general and $G = \sigma$ in the standard logit special case, i.e. when $\varepsilon_i^b - \varepsilon_i^a$ has the standard logistic distribution. The expected payoff to $A$ is:

$$Q_{\text{inst}}\big(\boldsymbol{\pi}^A, \boldsymbol{\pi}^B;\ \boldsymbol{\pi}^{A'}, \boldsymbol{\pi}^{B'}, \beth\big) =$$
$$\mathbb{E}_{\substack{\mathbf{t} \sim \bar{\boldsymbol{\pi}}^A(\boldsymbol{\pi}^A, \boldsymbol{\pi}^{A'}, \beth) \\ \mathbf{u} \sim \bar{\boldsymbol{\pi}}^B(\boldsymbol{\pi}^B, \boldsymbol{\pi}^{B'}, \beth)}}\left[V(\mathbf{t}, \mathbf{u})\right].$$

**Equilibria Under Institutions.** The minimax problem over interpretable, variance-controlled policies using factored, product-of-Categorical distributions, $\Pi_{\text{fact}}^A, \Pi_{\text{fact}}^B \subset \Delta(\mathcal{T})$ becomes (assuming fixed $\boldsymbol{\pi}^{A'}, \boldsymbol{\pi}^{B'}$):

$$\max_{\boldsymbol{\pi}^A} \min_{\boldsymbol{\pi}^B} Q_{\text{inst}}\big(\boldsymbol{\pi}^A, \boldsymbol{\pi}^B;\ \boldsymbol{\pi}^{A'}, \boldsymbol{\pi}^{B'}, \beth\big),$$

defining a *restricted minimax* problem. When the institutional pushforward (the mapping from strategy distributions to nominee distributions induced by the institutional rules) is affine in each player's mixed strategy with the opponent and counter-distributions held fixed, $Q_{\text{inst}}$ is bilinear on the full mixed-strategy simplices $\Delta(\mathcal{T}) \times \Delta(\mathcal{T})$. In that case, von Neumann's minimax theorem guarantees a full-simplex saddle point. Without this affine/bilinear condition, such an existence conclusion does not follow. Within the restricted factored class, $\Pi_{\text{fact}}^c$ is in general non-convex, so a saddle point need not exist within the restricted class. In practice, we compute a stationary point via gradient ascent–descent on unconstrained logits; see Appendix for payoff-based exploitability diagnostics that quantify the largest unilateral improvement available against the learned pair $(\widehat{\boldsymbol{\pi}}^A, \widehat{\boldsymbol{\pi}}^B)$.

**Guarantees.** The regularized payoff

$$\Phi(\pi^A, \pi^B) = Q_{\text{inst}}(\pi^A, \pi^B) - \lambda R(\pi^A \| p) + \lambda R(\pi^B \| p)$$

is concave in $\pi^A$ and convex in $\pi^B$ whenever $Q_{\text{inst}}$ is bilinear in the full mixed strategies, as under the affine primary

---

**Algorithm 1** Adversarial Minimax Policy Learning

---

**Require:** Forced-choice data $(C_i, \mathbf{T}_i^A, \mathbf{T}_i^B)_{i=1}^n$, institutional params $\beth$, reference policy $p$, regularization strength $\lambda$, regularizer $R$, learning rate $\gamma$, iterations $S$

    **Step 1: Outcome Model.** Fit $\hat{f}_\theta(\mathbf{t}^A, \mathbf{t}^B)$ via GLM/NN

    **Step 2: Restricted Nonconvex Optimization.**

    Initialize logits $\boldsymbol{\alpha}^{A,(0)}, \boldsymbol{\alpha}^{B,(0)}$

    Define $\Phi(\pi^A, \pi^B) = Q_{\text{inst}}(\pi^A, \pi^B; \beth) - \lambda R(\pi^A \| p) + \lambda R(\pi^B \| p)$.

    **for** $s = 1$ to $S$ **do**

        $\boldsymbol{\alpha}^{A,(s)} \leftarrow \boldsymbol{\alpha}^{A,(s-1)} + \gamma \nabla_{\boldsymbol{\alpha}^A} \Phi(\pi^A, \pi^B)$ {ascent}

        $\boldsymbol{\alpha}^{B,(s)} \leftarrow \boldsymbol{\alpha}^{B,(s-1)} - \gamma \nabla_{\boldsymbol{\alpha}^B} \Phi(\pi^A, \pi^B)$ {descent}

    **end for**

    $\hat{\pi}_{d\cdot}^{A*} \leftarrow \text{softmax}(\alpha_{d\cdot}^{A,(S)}), \hat{\pi}_{d\cdot}^{B*} \leftarrow \text{softmax}(\alpha_{d\cdot}^{B,(S)})$ {softmax within each factor $d$}

    **Step 3: Monte Carlo Nominee Estimation.** Compute $\hat{\bar{\boldsymbol{\pi}}}^{A*}, \hat{\bar{\boldsymbol{\pi}}}^{B*}$ via Monte Carlo over primary draws

    **Step 4: Uncertainty Quantification.**

    Compute Jacobian $\mathbf{J}$ via unrolling/implicit differentiation

    $\text{Var} \leftarrow \mathbf{J}\hat{\Sigma}\mathbf{J}'$ (Delta method)

    **Return:** $\hat{\boldsymbol{\pi}}^{A*}, \hat{\boldsymbol{\pi}}^{B*}, \widehat{Q}_{\text{inst}}, \widehat{\Phi}$, standard errors

---

pushforward conditions stated in the Appendix. In that convex–concave full-simplex setting, extragradient/Mirror-Prox methods converge with explicit primal–dual gap rates (Nemirovski, 2004). A factored logit parameterization is generally nonconvex, so the ascent–descent solver used below should be interpreted as computing a restricted stationary point, its quality assessed via exploitability diagnostics.

As in the average case, Delta-method inference follows by differentiating the resulting objective, either by backpropagating through the unrolled optimization trace or via implicit differentiation of the stationarity conditions at the fixed point, yielding standard errors for $\hat{\bar{\boldsymbol{\pi}}}^A, \hat{\bar{\boldsymbol{\pi}}}^B$ and $\widehat{Q}_{\text{inst}}$.

**Remarks.** *(i)* Open vs. closed primaries, heterogeneous turnouts, and the participation of independents are encoded by $\beth$ via the composition/weighting of $\mathcal{I}^A$, $\mathcal{I}^B$, and $\mathcal{E}$. *(ii)* Hard rules (eligibility constraints, ballot-access requirements) can be enforced by restricting the support of $\boldsymbol{\pi}^A$ and $\boldsymbol{\pi}^B$ to admissible profiles. *(iii)* Multi-round or multi-candidate primaries can be accommodated using the appropriate implied choice probabilities.

**Strategic Divergence.** Unlike AMCE analysis, which cannot quantify observed candidate information through experimental findings, the methodology here enables the measurement of strategic divergence using actual candidate profiles and the elicited conjoint preferences. In particular, given the optimal candidate distribution for one party, $\boldsymbol{\pi}^A$, and another, $\boldsymbol{\pi}^B$, in a given institutional context, we can find the strategic divergence factor, $\mathcal{D}_\varepsilon$, of a given candidate profile, $\mathbf{t}$, using the estimated strategies:

$$\mathcal{D}_\varepsilon(\mathbf{t}) = \left| \log\left( \frac{\text{Pr}_{\boldsymbol{\pi}^A}(\mathbf{t}) + \varepsilon}{\text{Pr}_{\boldsymbol{\pi}^B}(\mathbf{t}) + \varepsilon} \right) \right|, \qquad (4)$$

with a small $\varepsilon > 0$ to avoid undefined log ratios. When $\mathcal{D}_\varepsilon(\mathbf{t})$ is 0, the candidate profile $\mathbf{t}$ would be equally likely under the strategic action of party $A$ and $B$; when $\mathcal{D}_\varepsilon(\mathbf{t})$ is large, that profile would be likely under the strategy of one party, but unlikely under that of another.

# 4. Experiments

**Average Case Simulation.** In Monte Carlo simulations using synthetic binary conjoint data under a linear outcome model with interactions (scaled to $R^2 = 0.70$ for main effects), we assess finite-sample convergence of the average-case optimal stochastic intervention by varying sample sizes ($n \in \{500, 1500, 3500, 10000\}$) and dimensions ($K \in \{5, 10, 20\}$), with $L_2$ regularization tuned to diverge moderately from the uniform data-generating distribution. Results demonstrate negligible bias and rapidly declining RMSE (variance-dominated) for $\hat{\boldsymbol{\pi}}^*$ and $\widehat{Q}(\hat{\boldsymbol{\pi}}^*)$ even at small $n$, with inference reliable as coverage nears nominal levels across settings; details, including Figures 3–10, are in §B.4. This approach also substantially outperforms a baseline AMCE-based policy (selecting per-factor maximizers), achieving higher expected outcomes on average; see Fig. 4.

To assess sensitivity to outcome-model misspecification, we also evaluate settings where the true choice surface contains nonlinear structure while the fitted model is either a GLM with pairwise interactions or a Bayesian Transformer. Table 2 shows that the GLM is most efficient and best calibrated when the linear approximation is correct or close, while the Transformer can improve RMSE under stronger nonlinear misspecification but has imperfect coverage.

**Adversarial Case Simulation Design.** To assess finite-sample performance in the adversarial setting, we simulate two-party strategic competition between Republicans ($R$) and Democrats ($D$) in a two-stage electoral process: primaries for nominee selection, followed by a general election. Voters are affiliated with $R$ (fraction $p_R$) or $D$ ($p_D = 1 - p_R$), with only affiliated voters participating in their primary. We grid over $p_R \in \{0.2, 0.3, 0.5, 0.65, 0.8\}$ and sample sizes $n \in \{1000, 5000, 10000\}$, with Monte Carlo replications per cell.

In primaries, each party offers two profiles ($\mathbf{T}_i^{R,1}, \mathbf{T}_i^{R,2}$ for $R$; similarly for $D$), with one selected via the party's mechanism and the other uniform. Voter choices follow logistic models based on features (gender, for tractable ground-truth equilibria). General elections pit primary winners ($\mathbf{T}_i^{R,*}, \mathbf{T}_i^{D,*}$) against each other, with all voters choosing via separate $R$- and $D$-specific logistic models.

Ground-truth mixed strategies $\boldsymbol{\pi}^R, \boldsymbol{\pi}^D$ are computed as grid-approximated equilibria, maximizing $Q(\boldsymbol{\pi}^R, \boldsymbol{\pi}^D) = \mathbb{E}[\text{Pr}\{C_i(\mathbf{T}_i^R, \mathbf{T}_i^D) = 1\}]$ over a finite strategy grid. For each run, we estimate equilibria and outcomes, evaluating

how $p_R$ and $n$ affect performance. We report RMSE and 95% coverage for $\boldsymbol{\pi}^R$ (see §B.3).

**Adversarial Case Simulation Results.** Simulation results indicate that the estimation error depends primarily on the conjoint sample size, with only modest sensitivity to the proportion of Republican voters. Larger sample sizes reduce uncertainty by stabilizing the estimates of voter utilities: with larger sample sizes, the overall estimation error declines sharply for all values of $p_R$. Coverage rates fall below the nominal level for $n = 1000$ but approach the nominal 95% level for larger sample sizes. The stronger performance under increasing $n$ reflects the fact that voters' utilities are more precisely estimated, allowing us to obtain better approximations of the restricted minimax stationary point in a two-party adversarial competition.

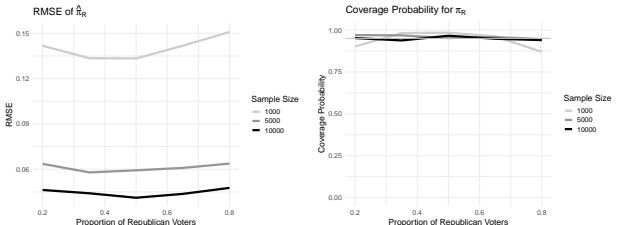

*Figure 1.* Finite-sample performance of $\widehat{\boldsymbol{\pi}}^R$ in the adversarial simulation. **Left:** Root-mean-squared error (RMSE) of $\widehat{\boldsymbol{\pi}}^R$ for different sample sizes and proportions of Republican voters, $p_R$. **Right:** Coverage probability of 95% confidence intervals for $\widehat{\boldsymbol{\pi}}^R$.

### 4.1. Real-World Conjoint Application: U.S. Presidential

We now apply our methods to analyze policy positioning and optimal candidate selection using presidential preference data from Ono & Burden (2019). Here, our outcome is a binary indicator stating whether candidate $a$ or $b$ was selected by respondent $i$ in a forced conjoint experiment. In the pairwise stochastic choice formulation above, this is the observed realization $C_i(\mathbf{T}_i^a, \mathbf{T}_i^b)$—an indicator of whether profile $\mathbf{T}_i^a$ is chosen over profile $\mathbf{T}_i^b$. Here, standard errors for Delta method uncertainty propagation are clustered at the respondent level. We report in the main text the neural approach and in the Appendix the GLM approach.

**Average vs. Adversarial Case Results.** We optimize expected vote share for subpopulations (all, Democrats, Republicans, independents) against a uniform opponent distribution, fitting an outcome model with interactions and propagating uncertainty via the Delta method. Optima (Fig. 13) diverge on immigration, abortion, and policy expertise (e.g., economy vs. public safety preferences), but converge on personality traits. Under closed primaries, we compute restricted minimax stationary points for Republican vs. Democrat strategies. Policies (Fig. 6) differ from average-case; e.g., Democrats deprioritize immigration in average case but counter Republican guest-worker stances adversarially.

Restricted-equilibrium vote shares drop markedly (Fig. 2), aligning closer to historical elections.

**Results with Data-Driven Clustering.** Prior analyses of regularized optimal stochastic interventions—with and without adversarial dynamics—ignored respondent characteristics beyond party affiliation. Yet, heterogeneous voter types often favor distinct candidate profiles. To uncover these differences, we apply optimal stochastic interventions under data-driven respondent clustering, revealing how subgroups respond uniquely to high-dimensional features. Leveraging the clustered outcome model of Goplerud et al. (2025), Fig. 12 shows that covariate-sensitive strategies recover the underlying Democrat-Independent-Republican preference structure endogenously, without explicit partisan labels. This highlights the approach's value in non-adversarial settings, where subgroup discovery enables tailored strategies.

**Historical Comparison.** In contrast to AMCEs, our methods yield *distributions* over profiles, enabling likelihood-based evaluation of observed candidates. We map the 2016 primary contenders to the conjoint levels of Ono & Burden (2019) (see §C.1); when a stance is ambiguous, we average uniformly over plausible levels. Fig. 2 shows that the average-case optimizer (uniform opponent) implies vote shares outside the historical two-party range since 1976, whereas the adversarial restricted-equilibrium optimizer closely matches the 2016 result and falls within the historical range. We then score each 2016 contender by the log probability of their features under the estimated optimal stochastic interventions. Fig. 11 aggregates the strategic divergence factor from Eq. 4; overall, Democratic candidates show somewhat higher divergence.

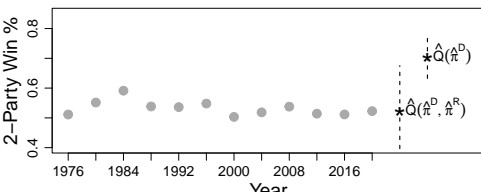

*Figure 2.* Comparing average-case and adversarial-case results with historical U.S. presidential election data; the adversarial restricted-equilibrium estimate falls inside this range. See Table 4 for candidate-level log-probabilities.

**Limitations.** This approach relies on a two-step estimator; inference requires accessible variance-covariance matrices. The factorized policy class is interpretable and variance-stabilizing, but is an approximation rather than an unrestricted optimum. Uncertainty estimates do not account for preference formation; inferred strategies depends on institutional design knowledge, which may be hard to obtain.

## Impact Statement

This work develops tools for transparent preference analysis, uncertainty-aware decision support, and reproducible mapping between observed candidates and conjoint feature spaces. The same tools could be misused for strategic persuasion, manipulative candidate positioning, or microtargeting. We therefore emphasize uncertainty reporting, public release of mappings and code, and diagnostics such as the strategic-divergence measure in Eq. 4, which can also help detect departures between observed candidates and inferred restricted-equilibrium policies.

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

# A. Theoretical Analysis

## A.1. The Optimal Stochastic Intervention in a Two-Way Interaction Model With Binary Factors

The objective function to maximize in the linear probability model formulation of the forced choice outcome is:

$$O(\boldsymbol{\pi}) = Q(\boldsymbol{\pi}) - \lambda||\mathbf{p} - \boldsymbol{\pi}||^2$$

$$= \beta_0 + \sum_{d=1}^{D} \beta_d \pi_d + \sum_{d'<d''} \gamma_{d',d''} \pi_{d'} \pi_{d''} - \lambda \sum_{d'''=1}^{D} \{(\pi_{d'''} - p_{d'''})^2 + ([1 - \pi_{d'''}] - [1 - p_{d'''}])^2\}$$

so that

$$\frac{dO}{d\pi_d} = \beta_d + \sum_{d'\neq d} \gamma_{d,d'} \pi_{d'} - 4\lambda(\pi_d - p_d) = 0$$

$$\implies$$

$$\sum_{d'\neq d} \gamma_{d,d'} \pi_{d'} - 4\lambda\pi_d = -\beta_d - 4\lambda p_d$$

where we use $\mathbf{p}_d$ to denote the vector of Categorical probabilities for all levels in factor $d$. This sets up a system of $D$ linear equations with $D$ unknowns, which can be represented in matrix form:

$$\mathbf{C}\boldsymbol{\pi}^* = \mathbf{B}$$
$$\boldsymbol{\pi}^* = \mathbf{C}^{-1}\mathbf{B},$$

where $B_{d,1} = -\beta_d - 4\lambda p_d$, $C_{d,d} = -4\lambda$ and $C_{d,d'} = \gamma_{d,d'}$. For $d \neq d'$, set $C_{d,d'} = C_{d',d} = \gamma_{d,d'}$.

## A.2. The Optimal Stochastic Intervention in a Two-Way Interaction Model With Multiple Factor Levels

The outcome model with multiple factor levels is

$$Y_i(t) = \beta_0 + \sum_{d=1}^{D} \sum_{l=1}^{L_d-1} \beta_{dl} t_{dl} + \sum_{d',d'':d'<d''} \sum_{l'=1}^{L_{d'}-1} \sum_{l''=1}^{L_{d''}-1} \gamma_{d'l',d''l''} \; t_{d'l'} \; t_{d''l''} + \epsilon_i,$$

where $t_{dl}$ denotes the binary indicator for whether level $l$ in factor $d$ is assigned. By linearity of expectations and independence of factors:

$$Q(\boldsymbol{\pi}) = \beta_0 + \sum_{d=1}^{D} \sum_{l=1}^{L_d-1} \beta_{dl} \pi_{dl} + \sum_{d',d'':d'<d''} \sum_{l'=1}^{L_{d'}-1} \sum_{l''=1}^{L_{d''}-1} \gamma_{d'l',d''l''} \; \pi_{d'l'} \; \pi_{d''l''}.$$

The objective is now

$$O(\boldsymbol{\pi}) = Q(\boldsymbol{\pi}) - \lambda||\mathbf{p} - \boldsymbol{\pi}||^2$$

$$= \beta_0 + \sum_{d=1}^{D} \sum_{l=1}^{L_d-1} \beta_{dl} \; \pi_{dl} + \sum_{d',d'':d'<d''} \sum_{l'=1}^{L_{d'}-1} \sum_{l''=1}^{L_{d''}-1} \gamma_{d'l',d''l''} \; \pi_{d'l'} \pi_{d''l''}$$

$$- \lambda \sum_{d'''=1}^{D} \left\{ \sum_{l'''=1}^{L_{d'''}-1} (\pi_{d'''l'''} - p_{d'''l'''})^2 + \left(1 - \left[ \sum_{l''''=1}^{L_{d'''}-1} \pi_{d'''l''''} \right] - \left(1 - \left[ \sum_{l''''=1}^{L_{d'''}-1} p_{d'''l''''} \right]\right)\right)^2 \right\}$$

so that, for $l < L_d$:

$$\frac{dO}{d\pi_{dl}} = \beta_{dl} + \sum_{d'\neq d} \sum_{l'=1}^{L_{d'}-1} \gamma_{dl,d'l'} \; \pi_{d'l'} - 2\lambda(\pi_{dl} - p_{dl}) - 2\lambda \left( \sum_{l'=1}^{L_d-1} (\pi_{dl'} - p_{dl'}) \right) = 0$$

$$\implies$$

$$\sum_{d'\neq d} \sum_{l'=1}^{L_{d'}-1} \gamma_{dl,d'l'} \; \pi_{d'l'} - 4\lambda\pi_{dl} - 2\lambda \sum_{l'\neq l, l'<L_d} \pi_{dl'} = -\beta_{dl} - 4\lambda p_{dl} - 2\lambda \sum_{l'\neq l, l'<L_d} p_{dl'}.$$

This again sets up a system of $\sum_{d=1}^{D}(L_d - 1)$ linear equations with the same number of unknowns, which can be represented in matrix form:

$$\mathbf{C}\boldsymbol{\pi}^* = \mathbf{B}$$
$$\boldsymbol{\pi}^* = \mathbf{C}^{-1}\mathbf{B}.$$

where, letting $r(\cdot)$ denote a function returning the appropriate index into the matrix rows/columns:

$$B_{r(dl),1} = -\beta_{dl} - 4\lambda p_{dl} - 2\lambda \sum_{l' \neq l, l' < L_d} p_{dl'}$$

$$C_{r(dl),r(dl)} = -4\lambda$$
$$C_{r(dl),r(dl')} = -2\lambda$$
$$C_{r(dl),r(d'l'')} = \gamma_{dl,d'l''}$$

For $d \neq d'$, set $C_{r(dl),r(d'l')} = C_{r(d'l'),r(dl)} = \gamma_{dl,d'l'}$.

### A.3. The Optimal Stochastic Intervention in a Two-Way Interaction Model Under Forced Choice Outcomes

Let $C_i = C_i(\mathbf{T}_i^a, \mathbf{T}_i^b)$ denote the observed pairwise forced-choice outcome from the stochastic choice potential defined in the main text. Under a linear-probability approximation,

$$\mathbb{E}[C_i \mid \mathbf{T}_i^a, \mathbf{T}_i^b] = \tilde{\mu} + \sum_{d,l} \beta_{dl}\{I_i^a(dl) - I_i^b(dl)\} + \sum_{d<d'} \sum_{l,l'} \gamma_{dl,d'l'}\{I_i^a(dl,d'l') - I_i^b(dl,d'l')\}.$$

Under antisymmetric coding and identical profiles, the symmetry condition sets the corresponding intercept to $1/2$; we write $\tilde{\mu}$ to allow generic baseline scenarios. There is no individual error term inside the conditional probability; an equivalent outcome-regression display would be $C_i = \mathbb{E}[C_i \mid \mathbf{T}_i^a, \mathbf{T}_i^b] + \epsilon_i$. Here $I_i^c(dl)$ and $I_i^c(dl,d'l')$ denote the corresponding main-effect and pairwise indicators for candidate $c$.

For the average-case optimizer, set the opponent policy to the reference distribution $\mathbf{p}$ and optimize over $\boldsymbol{\pi}^a$. The simplex constraint is handled by eliminating the baseline level:

$$\pi_{dL_d}^a = 1 - \sum_{l < L_d} \pi_{dl}^a.$$

Let $(\bar{\beta}, \bar{\gamma})$ be the baseline-coded coefficients defined by the same transformation used in Proposition 3.1. Up to terms that do not depend on $\boldsymbol{\pi}^a$, the $L_2$-regularized average-case objective is

$$O(\boldsymbol{\pi}^a; \mathbf{p}) = \sum_{d=1}^{D} \sum_{l < L_d} \bar{\beta}_{dl}\pi_{dl}^a + \sum_{d<d'} \sum_{l<L_d} \sum_{l'<L_{d'}} \bar{\gamma}_{dl,d'l'}\pi_{dl}^a \pi_{d'l'}^a$$
$$- \lambda \sum_{d=1}^{D} \left\{ \sum_{l<L_d} (\pi_{dl}^a - p_{dl})^2 + \left( \sum_{l<L_d} (\pi_{dl}^a - p_{dl}) \right)^2 \right\}.$$

Differentiating only after this baseline elimination gives, for each $d$ and $l < L_d$,

$$\sum_{d' \neq d} \sum_{l' < L_{d'}} \bar{\gamma}_{dl,d'l'}\pi_{d'l'}^a - 4\lambda\pi_{dl}^a - 2\lambda \sum_{\substack{m \neq l \\ m < L_d}} \pi_{dm}^a = -\bar{\beta}_{dl} - 4\lambda p_{dl} - 2\lambda \sum_{\substack{m \neq l \\ m < L_d}} p_{dm}.$$

This is the same baseline-eliminated linear system as Proposition 3.1, with the symmetric indexing of $\bar{\gamma}$ understood for $d' \neq d$.

### A.4. Restricted-Equilibrium Approximation via a Variational Principle

This section formalizes approximation guarantees for restricting policies to the product-of-Categorical (factored) class. The key observation is that, with KL (trust-region/entropy) regularization, the full-simplex optimizer is a Gibbs (log-linear) distribution, while the best factored policy is its mean-field variational approximation. We then bound the value gap in terms of interaction strength, $D$, $L$, and $\lambda$.

**Setup.** Let $\mathcal{T} = \prod_{d=1}^{D}[L_d]$ be the finite profile space. Let $p(\mathbf{t})$ be a *reference/logging* distribution with full support ($p(\mathbf{t}) > 0$ for all $\mathbf{t} \in \mathcal{T}$); in our application $p(\mathbf{t}) = \prod_d p_d(t_d)$ is factored. Let $u : \mathcal{T} \to \mathbb{R}$ denote a utility/payoff (e.g., expected win probability or a smooth surrogate).

For $\lambda > 0$, define the KL-regularized value over the full simplex:

$$V^{\star}(u) := \max_{\sigma \in \Delta(\mathcal{T})} \left\{ \mathbb{E}_{\mathbf{t} \sim \sigma}[u(\mathbf{t})] - \lambda \operatorname{KL}(\sigma \| p) \right\}. \tag{5}$$

Let $\Pi_{\text{fact}}$ be the factored (product) family: $\Pi_{\text{fact}} = \{\pi : \pi(\mathbf{t}) = \prod_{d=1}^{D} \pi_d(t_d), \ \pi_d \in \Delta([L_d])\}$. Define the restricted value

$$V^{\text{fact}}(u) := \max_{\pi \in \Pi_{\text{fact}}} \left\{ \mathbb{E}_{\mathbf{t} \sim \pi}[u(\mathbf{t})] - \lambda \operatorname{KL}(\pi \| p) \right\}. \tag{6}$$

Because $\Pi_{\text{fact}} \subset \Delta(\mathcal{T})$, we always have $V^{\text{fact}}(u) \leq V^{\star}(u)$.

**Lemma A.1** (Gibbs variational identity and optimality gap). *Let $\lambda > 0$ and $p$ have full support. Define $Z(u) := \sum_{\mathbf{t} \in \mathcal{T}} p(\mathbf{t}) \exp\{u(\mathbf{t})/\lambda\}$ and*

$$\sigma_u^{\star}(\mathbf{t}) := \frac{p(\mathbf{t}) \exp\{u(\mathbf{t})/\lambda\}}{Z(u)}.$$

*Then:*

1. *(Closed form value) $V^{\star}(u) = \lambda \log Z(u)$.*

2. *(Optimizer) $\sigma_u^{\star}$ uniquely maximizes Eq 5.*

3. *(Gap identity) For any $\sigma \in \Delta(\mathcal{T})$,*

$$V^{\star}(u) - \left( \mathbb{E}_{\sigma}[u] - \lambda \operatorname{KL}(\sigma \| p) \right) = \lambda \operatorname{KL}(\sigma \| \sigma_u^{\star}).$$

*Proof.* (1–2) Consider $\sigma_u^{\star}(\mathbf{t}) \propto p(\mathbf{t}) e^{u(\mathbf{t})/\lambda}$. Compute

$$\log \frac{\sigma_u^{\star}(\mathbf{t})}{p(\mathbf{t})} = \frac{u(\mathbf{t})}{\lambda} - \log Z(u).$$

Thus

$$\operatorname{KL}(\sigma_u^{\star} \| p) = \mathbb{E}_{\sigma_u^{\star}} \left[ \log \frac{\sigma_u^{\star}(\mathbf{t})}{p(\mathbf{t})} \right] = \frac{1}{\lambda} \mathbb{E}_{\sigma_u^{\star}}[u] - \log Z(u).$$

Plugging into the objective gives

$$\mathbb{E}_{\sigma_u^{\star}}[u] - \lambda \operatorname{KL}(\sigma_u^{\star} \| p) = \mathbb{E}_{\sigma_u^{\star}}[u] - \lambda \left( \frac{1}{\lambda} \mathbb{E}_{\sigma_u^{\star}}[u] - \log Z(u) \right) = \lambda \log Z(u),$$

so $V^{\star}(u) \geq \lambda \log Z(u)$.

For an arbitrary $\sigma$,

$$\operatorname{KL}(\sigma \| \sigma_u^{\star}) = \mathbb{E}_{\sigma} \left[ \log \frac{\sigma(\mathbf{t})}{\sigma_u^{\star}(\mathbf{t})} \right] = \mathbb{E}_{\sigma} \left[ \log \frac{\sigma(\mathbf{t})}{p(\mathbf{t})} \right] - \mathbb{E}_{\sigma} \left[ \log \frac{\sigma_u^{\star}(\mathbf{t})}{p(\mathbf{t})} \right] = \operatorname{KL}(\sigma \| p) - \frac{1}{\lambda} \mathbb{E}_{\sigma}[u] + \log Z(u),$$

since (using the definition of $\sigma_u^{\star}(\mathbf{t})$):

$$\log \frac{\sigma_u^{\star}(\mathbf{t})}{p(\mathbf{t})} = \frac{u(\mathbf{t})}{\lambda} - \log Z(u), \quad \text{so} \quad \mathbb{E}_{\sigma} \left[ \log \frac{\sigma_u^{\star}(\mathbf{t})}{p(\mathbf{t})} \right] = \frac{1}{\lambda} \mathbb{E}_{\sigma}[u] - \log Z(u).$$

Rearranging yields

$$\mathbb{E}_{\sigma}[u] - \lambda \operatorname{KL}(\sigma \| p) = \lambda \log Z(u) - \lambda \operatorname{KL}(\sigma \| \sigma_u^{\star}) \leq \lambda \log Z(u),$$

where the last inequality comes from the non-negativity of the KL term. This all implies $V^{\star}(u) \leq \lambda \log Z(u)$ and proves (1). The same display shows the objective is maximized iff $\operatorname{KL}(\sigma \| \sigma_u^{\star}) = 0$, i.e. $\sigma = \sigma_u^{\star}$, proving uniqueness in (2). The rearranged identity is exactly (3). $\qquad \square$

**Mean-Field Interpretation.** Lemma A.1(3) implies

$$V^{\text{fact}}(u) = V^{\star}(u) - \lambda \inf_{\pi \in \Pi_{\text{fact}}} \text{KL}(\pi \| \sigma_u^{\star}),$$

so the restricted optimizer is precisely the *reverse-KL projection* of the Gibbs distribution $\sigma_u^{\star}$ onto the product family—the classical mean-field variational approximation (Wainwright & Jordan, 2008).

**Interaction-Strength Model.** To obtain an explicit bound, we specialize to pairwise log-linear utilities. The quantities below are defined relative to a fixed pairwise representation of $u$. To avoid representation dependence, in applications, we use the canonical centered/ANOVA representation under the reference product measure $p$, so that main effects and pairwise interactions are uniquely determined by the usual zero-mean constraints. All interaction ranges $\Delta_{dd'}$ are computed from this fixed representation. Assume

$$u(\mathbf{t}) = \sum_{d=1}^{D} u_d(t_d) + \sum_{d<d'} u_{dd'}(t_d, t_{d'}). \tag{7}$$

Define the *per-edge* interaction range

$$\Delta_{dd'} := \max\left\{ \max_{a,a' \in [L_d]} \max_{b \in [L_{d'}]} \left| u_{dd'}(a,b) - u_{dd'}(a',b) \right|, \max_{a \in [L_d]} \max_{b,b' \in [L_{d'}]} \left| u_{dd'}(a,b) - u_{dd'}(a,b') \right| \right\}.$$

Define the *per-coordinate* bounded-differences constant

$$c_d := \sum_{d' \neq d} \Delta_{dd'}.$$

**Lemma A.2** (Bounded-differences MGF inequality). *Let $X = (X_1, \ldots, X_D)$ have independent coordinates and let $f(X)$ be any real function satisfying: for each $d$, for all $x, x'$ differing only in coordinate $d$, $|f(x) - f(x')| \leq c_d$. Then for all $s \in \mathbb{R}$,*

$$\log \mathbb{E}\left[ \exp\left( s(f(X) - \mathbb{E}f(X)) \right) \right] \leq \frac{s^2}{8} \sum_{d=1}^{D} c_d^2.$$

*Proof.* This is the standard bounded-differences (McDiarmid/Azuma-Hoeffding) MGF bound (McDiarmid, 1989). Define the Doob martingale $M_d := \mathbb{E}[f(X) \mid X_1, \ldots, X_d]$ so that $f(X) - \mathbb{E}f(X) = \sum_{d=1}^{D}(M_d - M_{d-1})$. Each increment $(M_d - M_{d-1})$ is almost surely bounded in an interval of width $c_d$ by the bounded-differences condition. Hoeffding's lemma bounds the conditional MGF of each increment by $\exp(s^2 c_d^2/8)$; iterating the tower property yields the stated bound. $\square$

**Theorem A.3** (Factored-policy approximation bound). *Assume $p(\mathbf{t}) = \prod_{d=1}^{D} p_d(t_d)$ and $u$ has the pairwise form (Eq. 7). Then for all $\lambda > 0$,*

$$0 \leq V^{\star}(u) - V^{\text{fact}}(u) \leq \frac{1}{8\lambda} \sum_{d=1}^{D} c_d^2 = \frac{1}{8\lambda} \sum_{d=1}^{D} \left( \sum_{d' \neq d} \Delta_{dd'} \right)^2. \tag{8}$$

*In particular, if $\Delta_{dd'} \leq \Delta_{\max}$ for all $d \neq d'$, then*

$$V^{\star}(u) - V^{\text{fact}}(u) \leq \frac{D(D-1)^2 \Delta_{\max}^2}{8\lambda}.$$

*Proof.* Write $u(\mathbf{t}) = a(\mathbf{t}) + b(\mathbf{t})$ where $a(\mathbf{t}) = \sum_d u_d(t_d)$ and $b(\mathbf{t}) = \sum_{d<d'} u_{dd'}(t_d, t_{d'})$.

Define the product distribution

$$q(\mathbf{t}) := \frac{p(\mathbf{t}) \exp\{a(\mathbf{t})/\lambda\}}{\sum_{\mathbf{u}} p(\mathbf{u}) \exp\{a(\mathbf{u})/\lambda\}}.$$

Because $p$ is product and $a$ is additive in coordinates, $q$ is also product, hence $q \in \Pi_{\text{fact}}$.

First, by Lemma A.1 applied to $a$,

$$\mathbb{E}_q[a] - \lambda \text{KL}(q \| p) = \lambda \log \sum_{\mathbf{t}} p(\mathbf{t}) e^{a(\mathbf{t})/\lambda}.$$

Therefore, evaluating the factored objective at $q$ gives

$$V^{\text{fact}}(u) \geq \Big( \mathbb{E}_q[a] - \lambda \text{KL}(q\|p) \Big) + \mathbb{E}_q[b] = \lambda \log \sum_{\mathbf{t}} p(\mathbf{t}) e^{a(\mathbf{t})/\lambda} + \mathbb{E}_q[b].$$

Second, the full value is

$$V^{\star}(u) = \lambda \log \sum_{\mathbf{t}} p(\mathbf{t}) e^{(a(\mathbf{t})+b(\mathbf{t}))/\lambda}$$

$$= \lambda \log \Bigg( \underbrace{\sum_{\mathbf{t}} p(\mathbf{t}) e^{a(\mathbf{t})/\lambda}}_{=:Z(a)} \cdot \mathbb{E}_{\mathbf{t}\sim q}\big[e^{b(\mathbf{t})/\lambda}\big] \Bigg)$$

$$= \lambda \log Z(a) + \lambda \log \mathbb{E}_q\big[e^{b(\mathbf{t})/\lambda}\big].$$

Subtracting the lower bound for $V^{\text{fact}}(u)$ yields

$$V^{\star}(u) - V^{\text{fact}}(u) \leq \lambda \log \mathbb{E}_q\big[e^{b(\mathbf{t})/\lambda}\big] - \mathbb{E}_q[b] = \lambda \log \mathbb{E}_q\Big[ \exp\Big( \frac{b(\mathbf{t}) - \mathbb{E}_q[b]}{\lambda} \Big) \Big].$$

Under $q$, coordinates $(t_1, \ldots, t_D)$ are independent. Moreover, changing coordinate $t_d$ affects $b(\mathbf{t})$ only through interaction terms incident to $d$, so for any fixed $t_{-d}$,

$$\sup_{t_d, t_d'} \big|b(t_d, t_{-d}) - b(t_d', t_{-d})\big| \leq \sum_{d' \neq d} \Delta_{dd'} = c_d.$$

Thus Lemma A.2 with $s = 1/\lambda$ gives

$$\log \mathbb{E}_q\Big[ \exp\Big( \frac{b(\mathbf{t}) - \mathbb{E}_q[b]}{\lambda} \Big) \Big] \leq \frac{1}{8\lambda^2} \sum_{d=1}^{D} c_d^2.$$

Multiplying by $\lambda$ yields Eq. 8. The simplified bound uses $c_d \leq (D-1)\Delta_{\max}$. $\qquad\square$

**From Value Gap to Restricted-Equilibrium Error.** In a KL-regularized two-player game, fixing the opponent reduces each player's regularized best-response problem to a one-player problem of the form Eq. 5. For player $A$, define the opponent-induced utility $u_{\pi^B}^A$ by

$$\Phi(\sigma, \pi^B) = \mathbb{E}_{\mathbf{t}\sim\sigma}\big[u_{\pi^B}^A(\mathbf{t})\big] - \lambda \text{KL}(\sigma\|p) + \text{const}(\pi^B).$$

For player $B$, equivalently write the minimization of $\Phi(\pi^A, \sigma)$ as maximization of $-\Phi(\pi^A, \sigma)$, and define $u_{\pi^A}^B$ by

$$-\Phi(\pi^A, \sigma) = \mathbb{E}_{\mathbf{u}\sim\sigma}\big[u_{\pi^A}^B(\mathbf{u})\big] - \lambda \text{KL}(\sigma\|p) + \text{const}(\pi^A).$$

Theorem A.3 bounds the additional best-response improvement from allowing an unrestricted full distribution over profiles (rather than a product distribution) only when these induced utilities admit the pairwise decomposition in Eq. 7 with finite interaction ranges. This condition is automatic in some pairwise log-linear settings, but it is not automatic for nonlinear logistic, neural, or institutionally pushed-forward objectives.

Define best-response values

$$\text{BR}_{\text{full}}^A(\pi^B) := \max_{\sigma\in\Delta(\mathcal{T})} \Phi(\sigma, \pi^B), \qquad \text{BR}_{\text{fact}}^A(\pi^B) := \max_{\pi\in\Pi_{\text{fact}}^A} \Phi(\pi, \pi^B),$$

and

$$\text{BR}_{\text{full}}^B(\pi^A) := \min_{\sigma\in\Delta(\mathcal{T})} \Phi(\pi^A, \sigma), \qquad \text{BR}_{\text{fact}}^B(\pi^A) := \min_{\pi\in\Pi_{\text{fact}}^B} \Phi(\pi^A, \pi).$$

**Corollary A.4** (Exploitability decomposition). *Let $\Phi(\cdot, \cdot)$ be a (regularized) zero-sum payoff for $A$ and suppose $(\pi^A, \pi^B) \in \Pi_{\mathrm{fact}}^A \times \Pi_{\mathrm{fact}}^B$. Define* full *exploitability components*

$$\epsilon_{\mathrm{full}}^A := \max_{\sigma \in \Delta(\mathcal{T})} \Phi(\sigma, \pi^B) - \Phi(\pi^A, \pi^B), \qquad \epsilon_{\mathrm{full}}^B := \Phi(\pi^A, \pi^B) - \min_{\sigma \in \Delta(\mathcal{T})} \Phi(\pi^A, \sigma),$$

*and the corresponding* restricted *(factored) exploitability components by replacing $\Delta(\mathcal{T})$ with $\Pi_{\mathrm{fact}}$: $\epsilon_{\mathrm{fact}}^A, \epsilon_{\mathrm{fact}}^B$. Then*

$$\epsilon_{\mathrm{full}}^A = \epsilon_{\mathrm{fact}}^A + \left( \mathrm{BR}_{\mathrm{full}}^A(\pi^B) - \mathrm{BR}_{\mathrm{fact}}^A(\pi^B) \right), \qquad \epsilon_{\mathrm{full}}^B = \epsilon_{\mathrm{fact}}^B + \left( \mathrm{BR}_{\mathrm{fact}}^B(\pi^A) - \mathrm{BR}_{\mathrm{full}}^B(\pi^A) \right),$$

*and hence*

$$\epsilon_{\mathrm{full}} \leq \epsilon_{\mathrm{fact}} + \max \left\{ \mathrm{BR}_{\mathrm{full}}^A(\pi^B) - \mathrm{BR}_{\mathrm{fact}}^A(\pi^B), \ \mathrm{BR}_{\mathrm{fact}}^B(\pi^A) - \mathrm{BR}_{\mathrm{full}}^B(\pi^A) \right\}.$$

*In particular, at a* restricted equilibrium *(a saddle point within $\Pi_{\mathrm{fact}}$), $\epsilon_{\mathrm{fact}} = 0$ and the full exploitability is controlled purely by the approximation terms.*

*Proof.* For $A$,

$$\epsilon_{\mathrm{full}}^A = \max_{\sigma \in \Delta(\mathcal{T})} \Phi(\sigma, \pi^B) - \Phi(\pi^A, \pi^B) = \underbrace{\max_{\pi \in \Pi_{\mathrm{fact}}} \Phi(\pi, \pi^B) - \Phi(\pi^A, \pi^B)}_{\epsilon_{\mathrm{fact}}^A} + \left( \max_{\sigma \in \Delta(\mathcal{T})} \Phi(\sigma, \pi^B) - \max_{\pi \in \Pi_{\mathrm{fact}}} \Phi(\pi, \pi^B) \right),$$

which is exactly the stated identity since the second parenthesis is the best-response gap. For $B$,

$$\epsilon_{\mathrm{full}}^B = \Phi(\pi^A, \pi^B) - \min_{\sigma \in \Delta(\mathcal{T})} \Phi(\pi^A, \sigma)$$

$$= \underbrace{\Phi(\pi^A, \pi^B) - \min_{\pi \in \Pi_{\mathrm{fact}}^B} \Phi(\pi^A, \pi)}_{\epsilon_{\mathrm{fact}}^B} + \left( \min_{\pi \in \Pi_{\mathrm{fact}}^B} \Phi(\pi^A, \pi) - \min_{\sigma \in \Delta(\mathcal{T})} \Phi(\pi^A, \sigma) \right),$$

which is the stated $B$-side approximation gap. Taking maxima yields the final inequality. $\qquad \square$

**Applying Theorem A.3 to the Approximation Terms.** The decomposition in Corollary A.4 is algebraic and does not require pairwise utilities. To upper bound the two best-response gaps using Theorem A.3, assume that $u_{\pi_B}^A$ and $u_{\pi_A}^B$ have pairwise decompositions as in Eq. 7. Let $\Delta_{dd'}^A(\pi^B)$ and $\Delta_{dd'}^B(\pi^A)$ denote their corresponding interaction ranges, and define

$$B_A(\pi^B) = \frac{1}{8\lambda} \sum_{d=1}^{D} \left( \sum_{d' \neq d} \Delta_{dd'}^A(\pi^B) \right)^2, \qquad B_B(\pi^A) = \frac{1}{8\lambda} \sum_{d=1}^{D} \left( \sum_{d' \neq d} \Delta_{dd'}^B(\pi^A) \right)^2.$$

Then

$$\mathrm{BR}_{\mathrm{full}}^A(\pi^B) - \mathrm{BR}_{\mathrm{fact}}^A(\pi^B) \leq B_A(\pi^B),$$

and

$$\mathrm{BR}_{\mathrm{fact}}^B(\pi^A) - \mathrm{BR}_{\mathrm{full}}^B(\pi^A) \leq B_B(\pi^A).$$

Consequently,

$$\epsilon_{\mathrm{full}} \leq \epsilon_{\mathrm{fact}} + \max\{B_A(\pi^B), B_B(\pi^A)\}.$$

For objectives that do not satisfy the pairwise induced-utility condition, the corollary still decomposes full exploitability into restricted exploitability plus approximation gaps, but these gaps must be computed, bounded by a separate argument, or reported as diagnostics.

**Remark.** The Gibbs/mean-field approximation result above is specific to KL regularization. For Euclidean penalties on the full joint simplex, even additive utilities need not yield a product-form optimizer. The closed-form $L_2$ result in Proposition 3.1 concerns a different restricted objective: we impose the product-of-Categoricals policy class and penalize deviations in factor marginals. We therefore do not claim a full-simplex mean-field approximation theorem for $L_2$ regularization.

## A.5. Convergence Guarantees for the Minimax Solver

This section states conditions under which the minimax problem is convex–concave in policy probabilities and provides convergence guarantees for standard first-order methods (extragradient / Mirror-Prox).

**Regularized Saddle-Point Problem.**  Let $\Pi^A, \Pi^B$ be convex compact sets (e.g., simplices or products of simplices). Consider a differentiable payoff $\Phi(\pi^A, \pi^B)$ for player $A$, where $A$ maximizes and $B$ minimizes:

$$\max_{\pi^A \in \Pi^A} \min_{\pi^B \in \Pi^B} \Phi(\pi^A, \pi^B).$$

In our application, $\Phi$ may include variance-control regularization (KL or $L_2$) so that the problem becomes well-conditioned.

**Assumption A.5** (Convex–concave smoothness).  $\Phi(\cdot, \cdot)$ is concave in $\pi^A$ for each fixed $\pi^B$ and convex in $\pi^B$ for each fixed $\pi^A$. For the Euclidean extragradient statement below, assume its gradient is $L$-Lipschitz on $\Pi^A \times \Pi^B$ in the Euclidean norm. This condition applies directly to smooth $L_2$-regularized objectives on the closed simplex. For KL regularization, either restrict the domain to a compact interior simplex $\Pi_\delta = \{\pi : \pi_j \geq \delta\}$ for some $\delta > 0$, or use the standard entropy-mirror/Mirror-Prox geometry on the relative interior rather than a Euclidean Lipschitz claim on the closed simplex.

**Theorem A.6** (Extragradient rate for convex–concave games). *Let $Z = \Pi^A \times \Pi^B$ be convex and compact, let $z = (\pi^A, \pi^B)$, and define*

$$F(z) = \left(-\nabla_{\pi^A} \Phi(\pi^A, \pi^B), \nabla_{\pi^B} \Phi(\pi^A, \pi^B)\right).$$

*Assume $F$ is monotone and $L$-Lipschitz on $Z$. Consider Euclidean extragradient*

$$y_t = \Pi_Z\{z_t - \eta F(z_t)\}, \qquad z_{t+1} = \Pi_Z\{z_t - \eta F(y_t)\},$$

*where $\Pi_Z$ denotes Euclidean projection, with $\eta \leq 1/(2L)$, and let $\bar{y}_T = T^{-1} \sum_{t=1}^{T} y_t$. Then*

$$\text{Gap}(\bar{y}_T) := \max_{\pi^A \in \Pi^A} \Phi(\pi^A, \bar{y}_T^B) - \min_{\pi^B \in \Pi^B} \Phi(\bar{y}_T^A, \pi^B) \ \leq \ \frac{1}{\eta T} \sup_{z \in Z} \|z - z_1\|_2^2 \leq \frac{\text{diam}(Z)^2}{\eta T}.$$

*For entropy geometry, the Euclidean squared diameter is replaced by the corresponding Bregman diameter.*

*Proof.* The convex–concave assumption implies monotonicity of $F$. The Euclidean extragradient one-step inequality gives, for every $z \in Z$,

$$\langle F(y_t), y_t - z \rangle \leq \frac{1}{2\eta}\{\|z_t - z\|_2^2 - \|z_{t+1} - z\|_2^2\},$$

up to the standard nonpositive Lipschitz residual controlled by $\eta \leq 1/(2L)$. Summing over $t = 1, \ldots, T$ and taking the supremum over $z \in Z$ yields

$$\sup_{z \in Z} \frac{1}{T} \sum_{t=1}^{T} \langle F(y_t), y_t - z \rangle \leq \frac{1}{2\eta T} \sup_{z \in Z} \|z - z_1\|_2^2.$$

By convexity/concavity, the left-hand variational-inequality residual upper bounds the primal–dual gap at the averaged iterate $\bar{y}_T$, giving the claim after absorbing constants according to the chosen extragradient normalization (Nemirovski, 2004). $\square$

**When is the Game Convex–Concave in Policy Probabilities?**  In the full mixed-strategy space, $Q_{\text{inst}}$ is linear in each player's joint distribution, hence convex–concave, when the institutional pushforward is affine in the deviating mixed strategy with opponent and counter-distributions held fixed. Adding a convex variance-control term (e.g., KL or $L_2$) preserves convex–concavity and can yield strong convexity/concavity.

In restricted (factored) policy parameterizations, convexity can fail in general. However, in the *quadratic* linear-probability + two-way-interactions regime, the Hessian in each player's probabilities is directly controlled by the interaction coefficients. If the regularization coefficient $\lambda$ dominates the relevant interaction Hessian terms, the regularized operator may become strongly monotone in the full-probability parameterization, in which case stronger convergence guarantees can be invoked under the corresponding assumptions (Nemirovski, 2004).

**Exploitability Certificate.** In convex–concave games, the primal–dual gap $\mathrm{Gap}(\bar{\pi}_T)$ upper bounds exploitability. In our institution-aware setting, we additionally compute the *external exploitability* defined later by exact full-simplex best responses (pure-profile maximizers), yielding an interpretable certificate even when restricted nonconvexities are present.

## A.6. Formal Asymptotics for the Learned Policy

We formalize the two-step estimator $(\hat{\pi}, \widehat{Q})$ using M-estimation and an implicit-function / Delta-method argument. The key points are: (i) consistency of the plug-in argmax (or saddle) mapping, (ii) differentiability ensured by interiority (entropy/KL regularization), and (iii) asymptotic normality via the Delta method.

**First Stage.** Let $\theta \in \Theta \subset \mathbb{R}^p$ be the outcome-model parameter (e.g., GLM coefficients or Bayesian neural network parameters). Let $\hat{\theta}$ be an estimator with

$$\sqrt{n}(\hat{\theta} - \theta_0) \Rightarrow \mathcal{N}(0, \Sigma_\theta). \tag{9}$$

**Second Stage (Average Case).** Let $x \in \mathcal{X} \subset \mathbb{R}^k$ be a free policy coordinate, for example baseline-eliminated probabilities with $\pi_{dL_d} = 1 - \sum_{l < L_d} \pi_{dl}$, or identifiable logits with one baseline level fixed per factor. Let $g : \mathcal{X} \to \Pi$ map free coordinates into the product of policy simplices, and define

$$m(x, \theta) := M(g(x), \theta).$$

The population optimizer is

$$x^\star(\theta) := \arg\max_{x \in \mathcal{X}} m(x, \theta), \qquad \pi^\star(\theta) := g(x^\star(\theta)), \qquad \hat{x} := x^\star(\hat{\theta}), \qquad \hat{\pi} := g(\hat{x}).$$

**Assumption A.7** (Interior uniqueness and smoothness in a free chart). There exists a neighborhood $\mathcal{N}$ of $\theta_0$ such that for all $\theta \in \mathcal{N}$: (i) $x^\star(\theta)$ is unique, (ii) $x^\star(\theta)$ lies in the interior of the free-coordinate domain $\mathcal{X}$, (iii) $m(x, \theta)$ is twice continuously differentiable in $x$ and continuously differentiable in $\theta$, and (iv) the Hessian

$$H_{xx}(\theta) := \nabla_x^2 m(x^\star(\theta), \theta)$$

is nonsingular, and negative definite in the average-case maximization problem.

**Theorem A.8** (Consistency and asymptotic normality of $\hat{\pi}$). *Under Eq. 9 and Assumption A.7, $\hat{\pi} \to_p \pi^\star(\theta_0)$ and*

$$J_x = -\Big[H_{xx}(\theta_0)\Big]^{-1} \nabla_{x\theta}^2 m(x^\star(\theta_0), \theta_0), \tag{10}$$

$$J_\pi = Dg(x^\star(\theta_0)) \, J_x.$$

*Consequently,*

$$\sqrt{n}\big(\hat{\pi} - \pi^\star(\theta_0)\big) \Rightarrow \mathcal{N}\Big(0, \; J_\pi \Sigma_\theta J_\pi^\top\Big).$$

*Proof. Step 1 (consistency).* Because $\hat{\theta} \to_p \theta_0$ and $M(\pi, \theta)$ is continuous in $(\pi, \theta)$ on a compact $\Pi$, $M(\pi, \hat{\theta}) \to_p M(\pi, \theta_0)$ uniformly in $\pi$. Uniqueness of $\pi^\star(\theta_0)$ implies $\hat{\pi} \to_p \pi^\star(\theta_0)$ by the argmax continuous mapping theorem (van der Vaart, 1998).

*Step 2 (differentiability of $x^\star(\theta)$).* The first-order condition in free coordinates is

$$\nabla_x m(x^\star(\theta), \theta) = 0.$$

Define $G(x, \theta) := \nabla_x m(x, \theta)$. Assumption A.7(iv) implies $\nabla_x G(x^\star(\theta_0), \theta_0) = H_{xx}(\theta_0)$ is nonsingular. The implicit function theorem therefore gives differentiability of $x^\star(\theta)$ near $\theta_0$ and

$$\nabla_\theta x^\star(\theta_0) = -\Big[H_{xx}(\theta_0)\Big]^{-1} \nabla_{x\theta}^2 m(x^\star(\theta_0), \theta_0).$$

The policy derivative follows by the chain rule: $J_\pi = Dg(x^\star(\theta_0)) J_x$.

*Step 3 (Delta method).* A first-order expansion gives $\hat{\pi} - \pi^\star(\theta_0) = J_\pi(\hat{\theta} - \theta_0) + o_p(n^{-1/2})$. Combine with equation 9 to obtain the stated asymptotic normality. $\square$

**Corollary A.9** (Asymptotics for the value $\widehat{Q}$)**.** *Let $V(\theta) := Q(\pi^\star(\theta); \theta)$ and $\hat{V} := Q(\hat{\pi}; \hat{\theta})$. Under the conditions of Theorem A.8 and differentiability of $Q$,*

$$\sqrt{n}(\hat{V} - V(\theta_0)) \Rightarrow \mathcal{N}(0, \; J_V \Sigma_\theta J_V^\top),$$

*with $J_V := \nabla_\theta V(\theta)|_{\theta=\theta_0}$ computable by the chain rule using $J_\pi$.*

**Minimax (Adversarial) Case.** Let $x = (x^A, x^B)$ denote free coordinates for both players and define $\phi(x^A, x^B, \theta) := \Phi(g_A(x^A), g_B(x^B), \theta)$. For an interior restricted saddle point, use the signed stationarity system

$$F(x, \theta) = \begin{pmatrix} \nabla_{x^A} \phi(x^A, x^B, \theta) \\ -\nabla_{x^B} \phi(x^A, x^B, \theta) \end{pmatrix} = 0.$$

If $\nabla_x F(x^\star, \theta_0)$ is nonsingular, the same implicit-function and Delta-method argument applies.

**Numerical Optimization and Simulation Error.** For numerical optimizers, the reported Delta method targets the exact stationary point provided the optimization residual is asymptotically negligible:

$$\|F(\hat{z}, \hat{\theta})\| = o_p(n^{-1/2}),$$

where $F$ denotes the relevant first-order/KKT mapping. If instead a fixed number $S$ of optimization steps is used, the Delta method applies to the finite-$S$ algorithmic estimand $z_S(\theta)$, not necessarily to the exact optimizer $z^\star(\theta)$. When the objective or its gradients are approximated by Monte Carlo with $M_n$ simulation draws and the simulation noise is omitted from the covariance, we require the simulation error to be $o_p(n^{-1/2})$, e.g., by exact summation or by taking $M_n$ large enough relative to $n$. Otherwise, the covariance must include an additional simulation-noise component or be interpreted conditional on fixed common random numbers.

**Remark.** When policies are allowed to lie on simplex boundaries, the argmax mapping can be non-differentiable (active-set changes). Entropy/KL regularization (or logit parameterization with finite logits) ensures interiority, making the implicit Jacobian equation 10 well-defined and aligning the formal theory with autodiff through the optimization routine.

**Certifiability.**

**Definition A.10** (Restricted equilibrium and unregularized external exploitability)**.** Let $\Pi_{\text{fact}}^A$, $\Pi_{\text{fact}}^B$ be the factored policy classes. A pair $(\pi^A, \pi^B) \in \Pi_{\text{fact}}^A \times \Pi_{\text{fact}}^B$ is a *restricted minimax equilibrium* if it is a saddle point of $Q_{\text{inst}}$ when each player is constrained to $\Pi_{\text{fact}}^c$. Given a candidate pair $(\pi^A, \pi^B)$, define

$$\epsilon_{\text{ext}}^A = \max_{\sigma \in \Delta(\mathcal{T})} Q_{\text{inst}}(\sigma, \pi^B; \pi^{A'}, \pi^{B'}, \beth) - Q_{\text{inst}}(\pi^A, \pi^B; \pi^{A'}, \pi^{B'}, \beth),$$

$$\epsilon_{\text{ext}}^B = Q_{\text{inst}}(\pi^A, \pi^B; \pi^{A'}, \pi^{B'}, \beth) - \min_{\sigma \in \Delta(\mathcal{T})} Q_{\text{inst}}(\pi^A, \sigma; \pi^{A'}, \pi^{B'}, \beth).$$

The *external exploitability* is $\epsilon_{\text{ext}} = \max\{\epsilon_{\text{ext}}^A, \epsilon_{\text{ext}}^B\}$. If $\epsilon_{\text{ext}} = 0$, then the pair is a full mixed-strategy equilibrium of the unregularized payoff game, assuming the best responses are computed exactly. More generally, $\epsilon_{\text{ext}} \le \epsilon$ means the pair is an $\epsilon$-Nash equilibrium in payoff: no player can improve expected vote share by more than $\epsilon$ through a unilateral deviation to any full mixed strategy. This is a payoff certificate and does not imply distributional closeness to a particular equilibrium without additional stability assumptions.

The external exploitability diagnostic above is intentionally defined using the unregularized institutional payoff $Q_{\text{inst}}$. It answers how much a player could gain in expected vote share by deviating to any full mixed strategy over profiles while holding the opponent fixed. It is therefore a substantive vote-share diagnostic, not the primal–dual gap of the regularized training objective $\Phi$. If one instead defines exploitability for the regularized game $\Phi$, the best-response problem includes the regularizer; KL regularization yields a Gibbs/interior best response, and $L_2$ regularization yields a projected affine best response, so the pure-profile certificate below does not apply.

Next, we show that, under the affine primary pushforward conditions stated below, the institution-aware payoff is affine in the focal player's joint distribution when the opponent and counter-distributions are fixed. The focal player's best response (over the full simplex) is then a pure profile found by maximizing a scalar score over profiles. This can be done by enumeration when $|\mathcal{T}|$ is manageable, or approximated via search/sampling/coordinate ascent for large $|\mathcal{T}|$, allowing us to compute exploitability without solving a complex inner optimization.

**Proposition A.11** (Pure best responses for affine unregularized deviations). *Consider the unregularized payoff $Q_{\text{inst}}$. Fix $\pi^{A'}, \pi^{B'}, \beth$, the opponent strategy $\pi^B$, and the induced opponent nominee distribution $\bar{\pi}^B$. Assume that, when A deviates, the A-side primary mechanism is represented by pairwise win probabilities $\kappa_A(\mathbf{t}, \mathbf{s})$ that depend only on the focal profile $\mathbf{t}$, the fixed counter-profile $\mathbf{s} \sim \pi^{A'}$, and institutional parameters, not on the full mixed strategy $\pi^A$ except through the outer mixture over $\mathbf{t}$. Then $Q_{\text{inst}}(\pi^A, \pi^B)$ is affine in $\pi^A$. Consequently, a full-simplex best response for A exists at a pure profile. The symmetric statement holds for B.*

*Proof.* Let

$$\Psi^B(\mathbf{v}) = \sum_{\mathbf{u}} \bar{\pi}^B(\mathbf{u}) V(\mathbf{v}, \mathbf{u})$$

be A's expected general-election value if A nominates profile $\mathbf{v}$, where

$$V(\mathbf{v}, \mathbf{u}) := \Pr\{C_i(\mathbf{v}, \mathbf{u}) = 1 \mid i \in \mathcal{E}\} = \mathbb{E}_{i \in \mathcal{E}}[G\{m_i(\mathbf{v}) - m_i(\mathbf{u})\}].$$

Under the pairwise primary rule, the A nominee distribution induced by a focal mixed strategy $\pi^A$ and a fixed counter-distribution $\pi^{A'}$ is

$$\bar{\pi}^A(\mathbf{v}) = \sum_{\mathbf{s}} \pi^{A'}(\mathbf{s}) \left[ \pi^A(\mathbf{v}) \kappa_A(\mathbf{v}, \mathbf{s}) + \mathbf{1}\{\mathbf{v} = \mathbf{s}\} \sum_{\mathbf{t}} \pi^A(\mathbf{t})\{1 - \kappa_A(\mathbf{t}, \mathbf{s})\} \right].$$

Therefore

$$Q_{\text{inst}}(\pi^A, \pi^B) = \sum_{\mathbf{v}} \bar{\pi}^A(\mathbf{v}) \Psi^B(\mathbf{v}) = \sum_{\mathbf{t}} \pi^A(\mathbf{t}) \sum_{\mathbf{s}} \pi^{A'}(\mathbf{s}) \left[ \kappa_A(\mathbf{t}, \mathbf{s}) \Psi^B(\mathbf{t}) + \{1 - \kappa_A(\mathbf{t}, \mathbf{s})\} \Psi^B(\mathbf{s}) \right].$$

Define the inner summand as $G^A(\mathbf{t})$. Then

$$Q_{\text{inst}}(\pi^A, \pi^B) = \sum_{\mathbf{t}} \pi^A(\mathbf{t}) G^A(\mathbf{t}),$$

which is affine in $\pi^A$. Maximizing a linear function over the simplex attains its maximum at an extreme point, so a pure profile $\mathbf{t}^\star \in \arg\max_{\mathbf{t}} G^A(\mathbf{t})$ is a best response. The exploitability component is

$$\epsilon^A_{\text{ext}} = \max_{\mathbf{t}} G^A(\mathbf{t}) - \sum_{\mathbf{t}} \pi^A(\mathbf{t}) G^A(\mathbf{t}).$$

$\square$

# B. Further Related Work

**Conjoint analysis methodology.** Conjoint experiments have become the dominant method for studying multidimensional preferences in political science and marketing (Hainmueller et al., 2014). Methodological advances include AMCE/AMIE estimation (Hainmueller et al., 2014; Egami & Imai, 2019), interpretation under marginalization choices (De la Cuesta et al., 2022; Abramson et al., 2022), design optimization (Bansak et al., 2018), hypothesis testing (Ham et al., 2024; Liu & Shiraito, 2023), and heterogeneity via clustered models (Goplerud et al., 2025). Our work shifts from effect estimation to *policy learning*—discovering optimal attribute distributions—while preserving interpretability through factored policies.

**Policy learning and treatment rules.** Extensive work addresses optimal treatment assignment from experimental and observational data (Dudik et al., 2011; Imai & Strauss, 2011; Zhao et al., 2012; Kitagawa & Tetenov, 2018; Athey & Wager, 2021; Ben-Michael et al., 2025; Kallus & Zhou, 2021; Zhang et al., 2022). These methods typically consider binary or low-dimensional treatments with individualized rules based on covariates. We extend this paradigm to combinatorial treatments (factorial profiles) with a *marginal* (attribute-level) policy class, enabling interpretable recommendations in high-dimensional action spaces.

**Game-theoretic learning.** Our adversarial extension connects to zero-sum games and equilibrium computation. Foundational work includes minimax solutions for matrix games (Von Neumann, 1928) and Markov games (Littman, 1994). Recent advances in multi-agent RL compute equilibria via fictitious play, policy gradient methods, or value iteration (Lanctot et al.,

2017). We adapt these ideas to a single-stage conjoint setting with product-of-Categoricals policies, tractable closed-form solutions under mild conditions, and formal uncertainty quantification for restricted-equilibrium strategies.

**Our contributions relative to prior work.** We are, to our knowledge, among the first to address optimal profile selection in conjoint experiments, particularly in adversarial settings. Key novelties include: (i) the factored stochastic policy estimand (not marginal effects); (ii) Delta-method uncertainty quantification for both $\pi^*$ and $Q(\pi^*)$; (iii) institution-aware minimax objectives embedding primary/general election structure; and (iv) a data-driven strategic divergence measure for comparing restricted-equilibrium polarization across parties.

## B.1. Methodological Details

**Covariate-Sensitive Strategies.** The approach discussed here can accommodate respondent covariates, as is possible in the sequential decision-making context (Lu et al., 2010). In our discussion up to now, the new treatment probabilities were assigned without considering the specific characteristics of each respondent. We could consider stochastic interventions that took into account covariate information in the targeting of the high-dimensional treatments:

$$Q(\pi^*) = \max_{\pi} Q(\pi) = \max_{\pi} \mathbb{E}_{\mathbf{X}} \left[ \mathbb{E}_{Y|\mathbf{X}} \left[ Y_i(\mathbf{t}) \mid \mathbf{X}_i = \mathbf{x} \right] \Pr_{\pi} \left( \mathbf{T}_i = \mathbf{t} \mid \mathbf{X}_i = \mathbf{x} \right) \right]$$

$$= \max_{\pi} \left\{ \sum_{\mathbf{x}} \sum_{\mathbf{t} \in \mathcal{T}} \mathbb{E}[Y_i(\mathbf{t}) \mid \mathbf{X}_i = \mathbf{x}] \Pr_{\pi} \left( \mathbf{T}_i = \mathbf{t} \mid \mathbf{X}_i = \mathbf{x} \right) \Pr\{\mathbf{X}_i = \mathbf{x}\} \right\}.$$

The covariate-sensitive distribution, $\Pr_{\pi}(\mathbf{T}_i \mid \mathbf{X}_i)$, can be operationalized by having different factor-level probabilities for each cluster, with a model predicting the cluster probabilities for each unit. If we let $\pi_{dlk}$ denote the probability of factor $d$, level $l$, for cluster $k \in \{1, ..., K\}$:

$$\Pr_{\pi}(\mathbf{T}_i \mid \mathbf{X}_i) = \sum_{k=1}^{K} \Pr_{\pi_k}(\mathbf{T}_i \mid Z_{ik} = 1) \; \Pr\{Z_{ik} = 1 | \mathbf{X}_i\}$$

$$= \sum_{k=1}^{K} \underbrace{\left\{ \prod_{d=1}^{D} \Pr_{\pi_k}(T_{id} | Z_{ik} = 1) \right\}}_{\text{Categorical probabilities for cluster } k} \underbrace{\Pr(Z_{ik} = 1 | \mathbf{X}_i = \mathbf{x})}_{\text{Softmax regression}} \tag{11}$$

In this context, estimation can be conducted using outcome models that cluster main and interaction effects (Goplerud et al., 2025).

## B.2. Simulation Details, Average Case

**Simulation Design: Average Case.** To probe finite-sample dynamics of the proposed optimal stochastic intervention methodologies for conjoint analysis, we employ Monte Carlo methods. In our simulations, we analyze synthetic factorial experiments with binary treatments where each treatment is drawn from an independent Bernoulli with probability parameter 0.5. We adopt a linear outcome model with interactions[1]:

$$Y_i(\mathbf{T}_i) = \beta_0 + \sum_{d=1}^{D} \sum_{l=1}^{L_d-1} \beta'_{dl} \, \mathbb{I}\{T_{id} = l\} + \sum_{d',d'':d'<d''} \sum_{l'=1}^{L_{d'}-1} \sum_{l''=1}^{L_{d''}-1} \gamma_{d',d''} \, \mathbb{I}\{T_{id'} = l'\} \mathbb{I}\{T_{id''} = l''\} + \epsilon_i,$$

with $\epsilon_i \sim N(0, 0.1)$, since this makes the computation of $Q(\theta)$ straightforward (in particular, $Q(\pi) = \beta' \pi + \sum_{d,d':d<d'} \gamma_{d,d'} \pi_d \pi_{d'}$). The coefficients are drawn i.i.d. from $N(0,1)$, and the interaction coefficients are scaled so that the $R^2$ in using the main effects only to predict the outcome is 0.70 (ensuring some effective nonlinearity). We obtain the true value of $\pi^*$ fixing $\lambda$ and solving for $\pi^*$ using Proposition 3.1.

To analyze finite sample convergence of $\hat{\pi}^*$, we vary the number of observations, $n \in \{500, 1500, 3500, 10000\}$. To analyze performance in the high-dimensional setting, where the number of treatment combinations is greater than the number of observations, we vary the number of factors, $K \in \{5, 10, 20\}$. We fix $\lambda$ so that the regularized optimal

---

[1]For simplicity, we here do not adopt the sum-to-0 coefficient constraint, and instead use a baseline category.

*Table 1.* Comparing different approaches to conjoint analysis. Pr here refers to the data-generating probability distribution over candidate features; $\text{Pr}_\pi$ refers to the distribution defining an optimal stochastic intervention. SI denotes "stochastic intervention"; GLM denotes "generalized linear model."

| | Average Marginal Component Effect (AMCE) | Average Marginal Interaction Effect (AMIE) | Average Case Optimal Stochastic Intervention | Adversarial Case Optimal Stochastic Intervention |
|---|---|---|---|---|
| *Character* | | | | |
| Components considered at a time | 1 | 2+ | All | All |
| Baseline factor category specified? | Yes | Yes | No | No |
| Marginalization over: | Respondents; other factors of reference profile via Pr; all factors of opponent profile via Pr | Respondents; other factors of reference profile via Pr; all factors of opponent profile via Pr | Respondents; factors of reference via $\text{Pr}_\pi$, opponent profile via Pr | Respondents; factors of reference profile via $\text{Pr}_{\pi^a}$, opponent profile via $\text{Pr}_{\pi^b}$ |
| Informative about strategy in an adversarial setting? | No | No | No | Yes |
| Hyper-parameters | Strength of regularization in outcome model (rarely used) | Strength of regularization in outcome model if used | Strength of regularization in outcome model; SI regularization | Strength of regularization in outcome model; SI regularization |
| Uncertainty estimation | GLM variance-covariance; bootstrap | GLM variance-covariance; bootstrap | GLM variance-covariance + Delta method | GLM variance-covariance + Delta method |
| *Data Requirements* | | | | |
| Requires forced-choice design? | No | No | No | Yes |
| Requires distinct respondent and profile sub-groups? | No | No | No | Yes |

stochastic interventions have no factor probabilities greater than 0.9, while having a degree of divergence from the (uniform) data-generating probabilities.

For the nonlinear misspecification robustness analysis summarized in the main text, we use the same sample-size and dimension grid and vary the misspecification magnitude. Table 3 reports the disaggregated value and policy recovery metrics by outcome model, misspecification level, and $K$.

**Simulation Results: Average Case.** First, we examine the degree to which $\widehat{\pi}^*$, the optimal stochastic intervention factor probabilities, and $\widehat{Q}(\widehat{\pi}^*)$, the average outcome under the regularized stochastic intervention, converge to the true values as the sample size grows. We see in the left panel of Fig. 3 that, with a small number of factors (5), the bias of $\widehat{\pi}^*$ is insignificant even with a small sample size (500). The variance of estimation contributes more prominently to the overall RMSE for all numbers of covariates; the variance decreases rapidly with the sample size. We see a similar pattern for $\widehat{Q}(\widehat{\pi}^*)$ in right panel of Fig. 3, where the bias is nominal with a small number of factors and the variance contributes more prominently to the overall RMSE, which still decreases with the sample size. Results are consistent with the idea that the optimal stochastic interventions are more difficult to estimate if there are more candidate features involved.

We next compare the value of the optimal stochastic intervention, $Q(\widehat{\pi}^*)$, against a simple baseline policy that, for each factor, places all mass on the level with the highest estimated main effect from a main-effects-only model (i.e., a degenerate

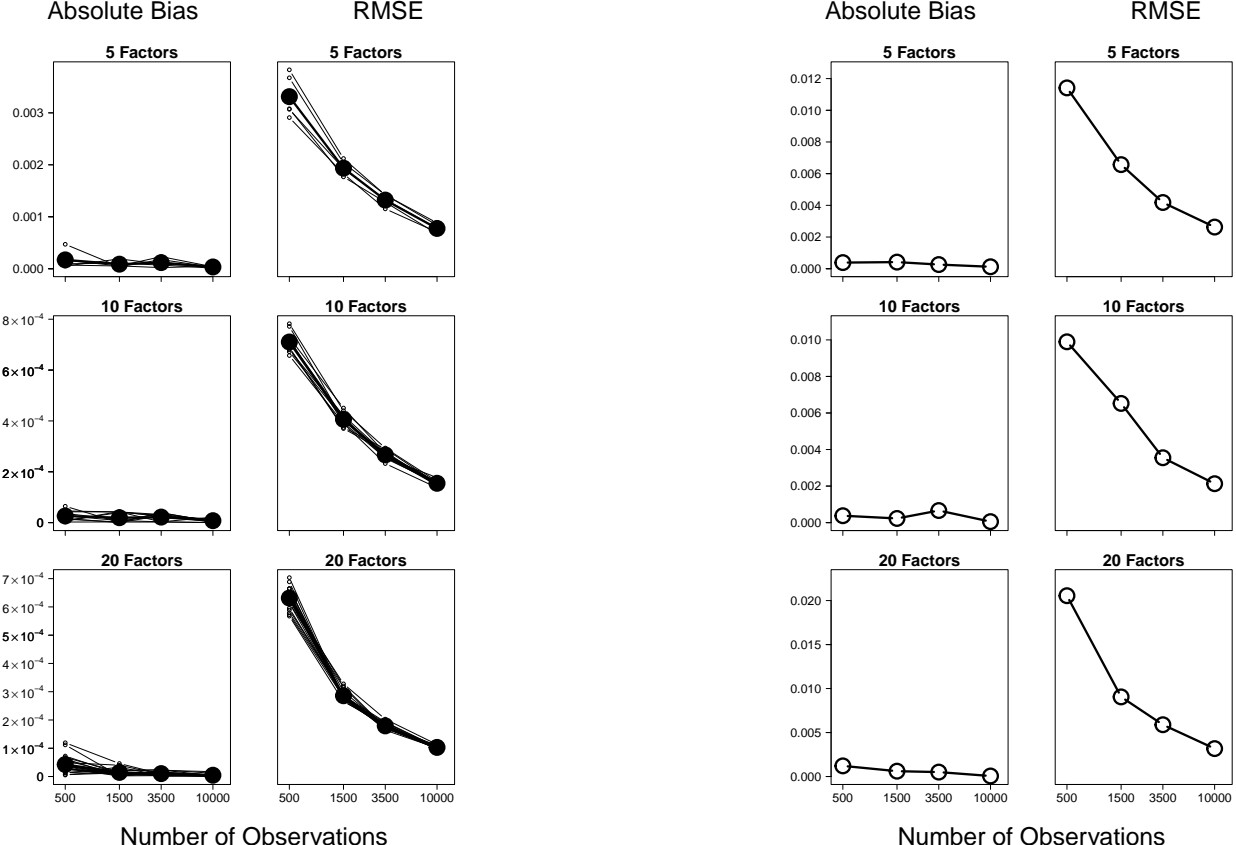

*Figure 3.* **Left:** Estimation bias and RMSE of $\boldsymbol{\pi}^*$. Each line represents one entry in $\boldsymbol{\pi}^*$. The bold line and closed circles represent the average value. **Right:** The estimation bias and RMSE of $Q(\boldsymbol{\pi}^*)$.

policy selecting per-factor AMCE maximizers, ignoring interactions). As shown in Fig. 4, the optimal SI method yields a higher mean value of 1.550 compared to the Max-AMCE baseline's 0.783, demonstrating the benefits of accounting for interactions and using stochastic rather than deterministic policies. The improvement is particularly pronounced in higher dimensions ($D = 20$ factors), where the complex interaction structure makes the main-effects-only approximation most inadequate.

We next consider estimated uncertainties compared against true sampling uncertainties. We see in Fig. 7 that the asymptotic variance of $\widehat{Q}(\hat{\boldsymbol{\pi}}^*)$ is somewhat underestimated for small sample sizes. Fig. 8 in §B.4 reports the true sampling variability of $\hat{\boldsymbol{\pi}}^*$ against the average standard error estimate from asymptotic inference; estimates are neither systematically too wide nor too narrow. Finally, we examine coverage, which combines information about point with variance estimates. We see in Fig. 9 coverage close to the target coverage rate across the number of factors and observations for the components of $\boldsymbol{\pi}^*$ and (in Fig. 10) for $\widehat{Q}(\hat{\boldsymbol{\pi}}^*)$ itself.

### B.3. Simulation Design: Adversarial Case

In order to investigate finite-sample performance under the more complex adversarial setting, we simulate strategic behavior between two hypothetical political parties denoted by $R$ (Republican) and $D$ (Democrat). We design a two-stage electoral process in which each party first selects a nominee via a primary election, and then those nominees compete in a general election. Voters differ by party affiliation, which determines whether they participate in the corresponding primary. Let $p_R$ denote the fraction of Republican voters in the electorate, so that $p_D = 1 - p_R$ is the fraction of Democratic voters. For each simulation run, we fix $p_R \in \{0.2, 0.3, 0.5, 0.65, 0.8\}$ along a grid, and we vary the conjoint sample size $n \in \{1000, 5000, 10000\}$. Each grid cell is replicated across Monte Carlo draws.

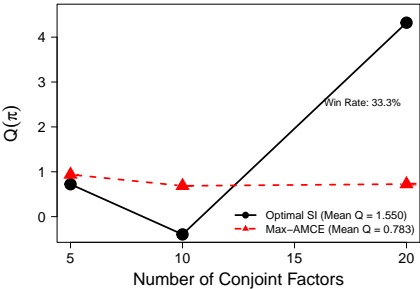

*Figure 4.* Expected outcome $Q(\boldsymbol{\pi})$ under the optimal stochastic intervention vs. the Max-AMCE baseline (deterministic policy selecting per-factor AMCE maximizers). Error bars show 95% confidence intervals. The improvement from optimal SI is largest at $D = 20$ factors, where interaction effects dominate.

Within each simulated dataset, we generate responses for primary and general-election stages. In the first stage, only voters from party $R$ or party $D$ participate in their own party's primary. We assign two potential candidate profiles for party $R$ and two for party $D$; one of these candidates is selected using the party's assignment mechanism, the other uniformly. Let these be $\mathbf{T}_i^{R,1}, \mathbf{T}_i^{R,2}$ for $R$ and $\mathbf{T}_i^{D,1}, \mathbf{T}_i^{D,2}$ for $D$. We specify probabilities with which each candidate profile is chosen by each respondent in that primary, using logistic models to capture how voters respond to candidate features—here, simply gender for tractability when computing ground truth equilibria via dense grid search. In the second stage, *all* voters, $R$ and $D$, face a forced choice in the general election between $\mathbf{T}_i^{R,*}$ and $\mathbf{T}_i^{D,*}$, winners of the respective primaries. In the second stage, Republican and Democrat voters select candidates again using two logistic models.

Having outlined the data-generating process, we now discuss how we compute the ground-truth strategies approximating a Nash equilibrium in the space of possible profile distributions for each party. The quantities $\boldsymbol{\pi}^R$ and $\boldsymbol{\pi}^D$ describe mixed strategies over candidate characteristics for $R$ and $D$, respectively. We define

$$Q(\boldsymbol{\pi}^R, \boldsymbol{\pi}^D) \;=\; \mathbb{E}_{\mathbf{T}_i^R \sim \boldsymbol{\pi}^R,\, \mathbf{T}_i^D \sim \boldsymbol{\pi}^D}\Big[\Pr\{C_i(\mathbf{T}_i^R, \mathbf{T}_i^D) = 1\}\Big],$$

where $\mathbf{T}_i^R$ and $\mathbf{T}_i^D$ represent each party's selection (who competes against the primary challenger). We compute grid-approximated equilibrium targets by evaluating each party's best response over finite grids $\mathcal{G}_R$ and $\mathcal{G}_D$ of mixed strategies.

In order to evaluate the finite-sample performance of the proposed algorithm for the adversarial setting, we implemented the two-stage design described in the preceding section while varying the proportion of Republican voters, $p_R$, in the electorate. That is, for each Monte Carlo run, we save the estimated equilibrium distribution for $\boldsymbol{\pi}^R$ and $\boldsymbol{\pi}^D$, along with the realized general-election outcomes under those strategies. By aggregating results across the grid of $\{p_R, n_{\mathrm{obs}}\}$ and across replications, we examine trends in how party composition $p_R$ and sample size $n_{\mathrm{obs}}$ affect equilibrium strategies, estimated vote shares, and convergence. This design allows us to evaluate the proposed adversarial methodology under changing population compositions and sample sizes. We focus on summarizing estimation accuracy for $\boldsymbol{\pi}^R$: we record root-mean-squared error (RMSE) and coverage of confidence intervals under repeated sampling, with coverage targeting the nominal rate of 95%.

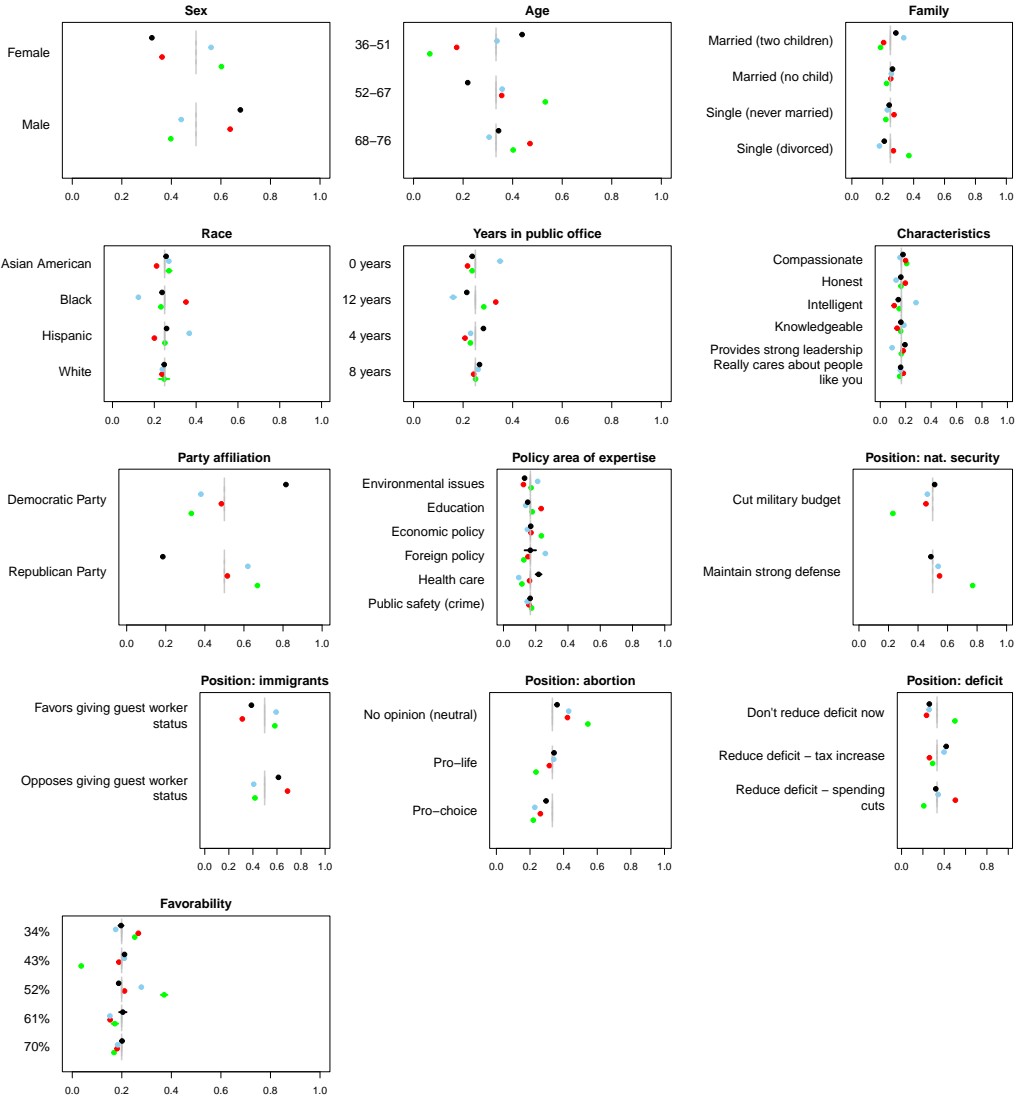

*Figure 5.* Optimal strategies in the average case setting. Black, blue, red, and green denote the average case optimal among all, Democrat, Republican, and Independent respondents in the sample.

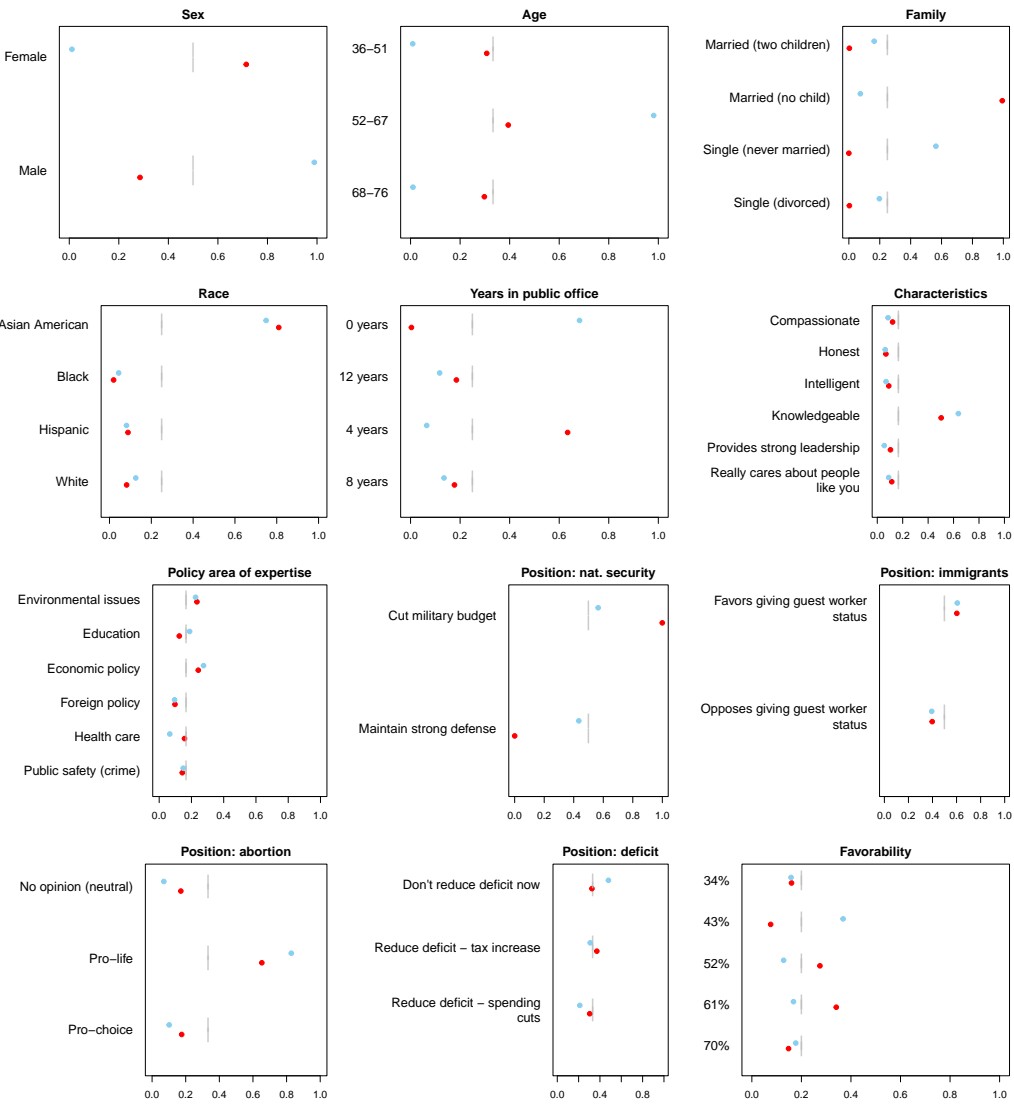

*Figure 6.* Optimal strategies in an adversarial setting. Blue/red denote the restricted-equilibrium strategy for the agent facing Democratic/Republican voters in the primary stage, respectively.

## B.4. Average Case Simulation Results

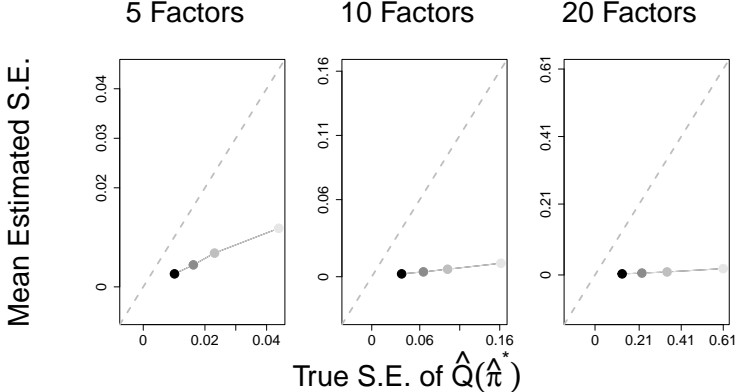

*Figure 7.* Points depict the average estimated standard deviation obtained via the Delta method. Colors depict the sample size (with $n = 500$ being light gray and $n = 10{,}000$ being black).

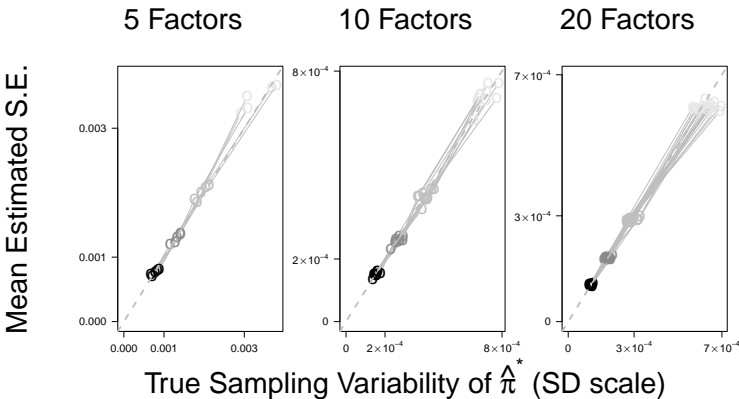

*Figure 8.* True sampling variability of $\hat{\boldsymbol{\pi}}^*$ plotted against the variability estimated via asymptotic inference. Colors depict the sample size (with $n$=500 being light gray and $n$=10,000 being black).

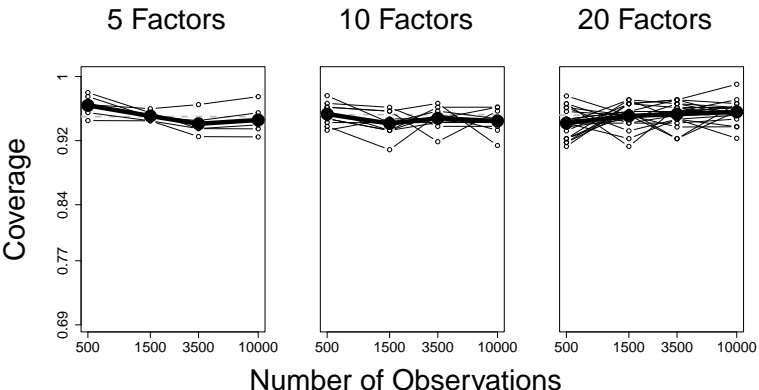

*Figure 9.* Finite-sample coverage of $\boldsymbol{\pi}^*$. Each line represents one entry in $\boldsymbol{\pi}^*$. The bold line and closed circles represent the average value.

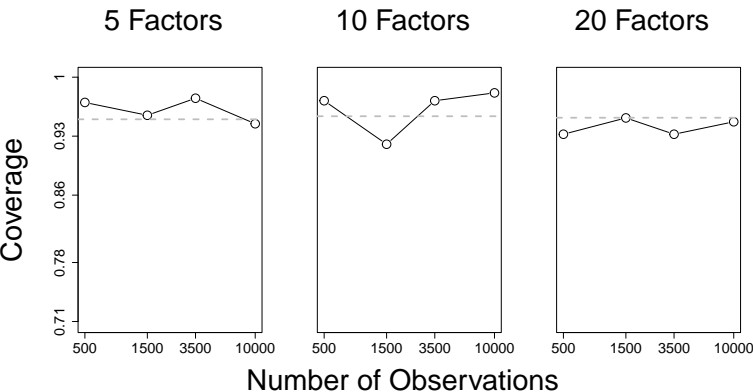

*Figure 10.* Finite-sample coverage for $Q(\pi^*)$.

*Table 2.* Robustness analysis under outcome-model misspecification. The true choice model includes nonlinear structure while the fitted model uses either a GLM with pairwise interactions or a Bayesian Transformer outcome model. Values are averaged over $n \in \{500, 1500, 3500, 10000\}$ and $K \in \{5, 10, 20\}$ simulation cells.

| Outcome model | Misspec. | Coverage for $Q$ | RMSE for $Q$ |
|---|---|---|---|
| Pairwise GLM | 0.0 | 0.94 | 0.007 |
| Pairwise GLM | 0.1 | 0.66 | 0.158 |
| Pairwise GLM | 0.5 | 0.78 | 0.553 |
| Transformer | 0.0 | 0.43 | 0.311 |
| Transformer | 0.1 | 0.56 | 0.216 |
| Transformer | 0.5 | 0.50 | 0.424 |

*Table 3.* Disaggregated misspecification robustness results by outcome model, misspecification magnitude, and number of factors $K$. Coverage and RMSE for $Q$ summarize value recovery; coverage and RMSE for $\pi^*$ summarize policy recovery.

| Outcome model | Misspec. | $K$ | $Q$ coverage | $Q$ RMSE | $\pi^*$ coverage | $\pi^*$ RMSE |
|---|---|---|---|---|---|---|
| Bayesian Transformer | 0.0 | 5 | 0.988 | 0.299 | 0.758 | 0.028 |
| Bayesian Transformer | 0.0 | 10 | 0.882 | 0.682 | 0.340 | 0.025 |
| Bayesian Transformer | 0.0 | 20 | 0.650 | 1.319 | 0.000 | 0.017 |
| Bayesian Transformer | 0.1 | 5 | 1.000 | 0.299 | 0.718 | 0.031 |
| Bayesian Transformer | 0.1 | 10 | 0.850 | 0.716 | 0.326 | 0.028 |
| Bayesian Transformer | 0.1 | 20 | 0.632 | 1.394 | 0.000 | 0.019 |
| Bayesian Transformer | 0.5 | 5 | 0.862 | 0.568 | 0.510 | 0.132 |
| Bayesian Transformer | 0.5 | 10 | 0.924 | 0.639 | 0.343 | 0.034 |
| Bayesian Transformer | 0.5 | 20 | 0.612 | 1.508 | 0.000 | 0.023 |
| GLM with pairwise interactions | 0.0 | 5 | 0.938 | 0.007 | 0.955 | 0.002 |
| GLM with pairwise interactions | 0.0 | 10 | 0.950 | 0.006 | 0.951 | 0.000 |
| GLM with pairwise interactions | 0.0 | 20 | 0.937 | 0.010 | 0.950 | 0.000 |
| GLM with pairwise interactions | 0.1 | 5 | 0.425 | 0.050 | 0.508 | 0.017 |
| GLM with pairwise interactions | 0.1 | 10 | 0.875 | 0.089 | 0.776 | 0.007 |
| GLM with pairwise interactions | 0.1 | 20 | 0.688 | 0.335 | 0.687 | 0.010 |
| GLM with pairwise interactions | 0.5 | 5 | 0.550 | 0.510 | 0.462 | 0.127 |
| GLM with pairwise interactions | 0.5 | 10 | 0.925 | 0.231 | 0.815 | 0.019 |
| GLM with pairwise interactions | 0.5 | 20 | 0.875 | 0.917 | 0.862 | 0.023 |

# C. Application Results

## C.1. Mapping the 2016 Candidate Primary Features onto the Conjoint Levels of Ono & Burden (2019)

We map the features of the 2016 presidential election candidates onto the conjoint features of Ono & Burden (2019). In some cases, this mapping is straightforward (e.g., with candidate gender). In other cases, the mapping is less straightforward. For example, the factor levels associated with marital status do not encompass the full range of possibilities seen among 2016 candidates. In such cases, we select the closest mapping (see Replication Data for full details). For example, a real, married candidate with 4 children would be mapped to the "Married with 2 children" level (not the "Single, divorced" or"Married, no children" levels).

We study these substantive questions by integrating the experiment mentioned above from Ono & Burden (2019). In this election, 17 Republican and 6 Democratic candidates vied for their respective parties' nominations in primaries. These candidates have a large number of features, which we mapped onto the conjoint factors of Ono & Burden (2019) (see §C.1 for details). Below we present this mapping for four of the candidates:

* *Ben Carson:* Republican, Black, male, 68-76, married (with children), 0 years of political experience, compassionate, policy focus on health care, emphasis on maintaining strong defense, opposes giving guest worker status, pro-life, don't reduce deficit now.

* *Hillary Clinton:* Democrat, White, female, 68-76, married (with children), 16 years of political experience, provides strong leadership, foreign policy, maintains strong defense, favors giving guest worker status, pro-choice, don't reduce deficit now.

* *Bernie Sanders:* Democrat, White, male, 68-76, married (with children), 34 years of political experience, compassionate, policy focus on economy, cut military budget, ambiguous position on immigration, pro-choice, reduce deficit through tax increase.

* *Donald Trump:* Republican, White, male, 68-76, married (with children), 0 years of political experience, provides strong leadership, policy focus on economy, emphasis on maintaining strong defense, opposes giving guest worker status, pro-life, reduce deficit through spending cuts.

## C.2. Application Results - Outcome Model (neural)

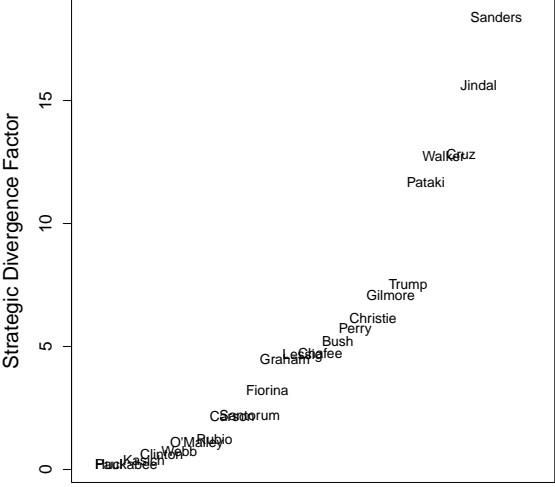

*Figure 11.* Strategic divergence factor computed for major candidates in the 2016 primaries (neural outcome model).

*Table 4.* Mean log-probabilities of 2016 primary candidates under estimated optimal stochastic interventions (neural outcome model). Higher values indicate profiles more consistent with restricted-equilibrium strategies. The strategic log-odds ratio $\mathcal{D}_\varepsilon(\mathbf{t}) = \left| \log\{(\mathrm{Pr}_{\boldsymbol{\pi}_A}(\mathbf{t}) + \varepsilon)/(\mathrm{Pr}_{\boldsymbol{\pi}_B}(\mathbf{t}) + \varepsilon)\} \right|$ quantifies divergence from Eq. 4. Standard errors reflect uncertainty propagated via the Delta method from the neural outcome model.

| Party | Quantity | Mean Log Prob. (s.e.) |
|---|---|---|
| Democrats | Average case | -14.92 (0.56) |
| Democrats | Adversarial case | -23.33 (1.41) |
| Democrats | Log likelihood ratio | -8.41 |
| Republicans | Average case | -17.29 (0.62) |
| Republicans | Adversarial case | -24.29 (0.76) |
| Republicans | Log likelihood ratio | -7.01 |

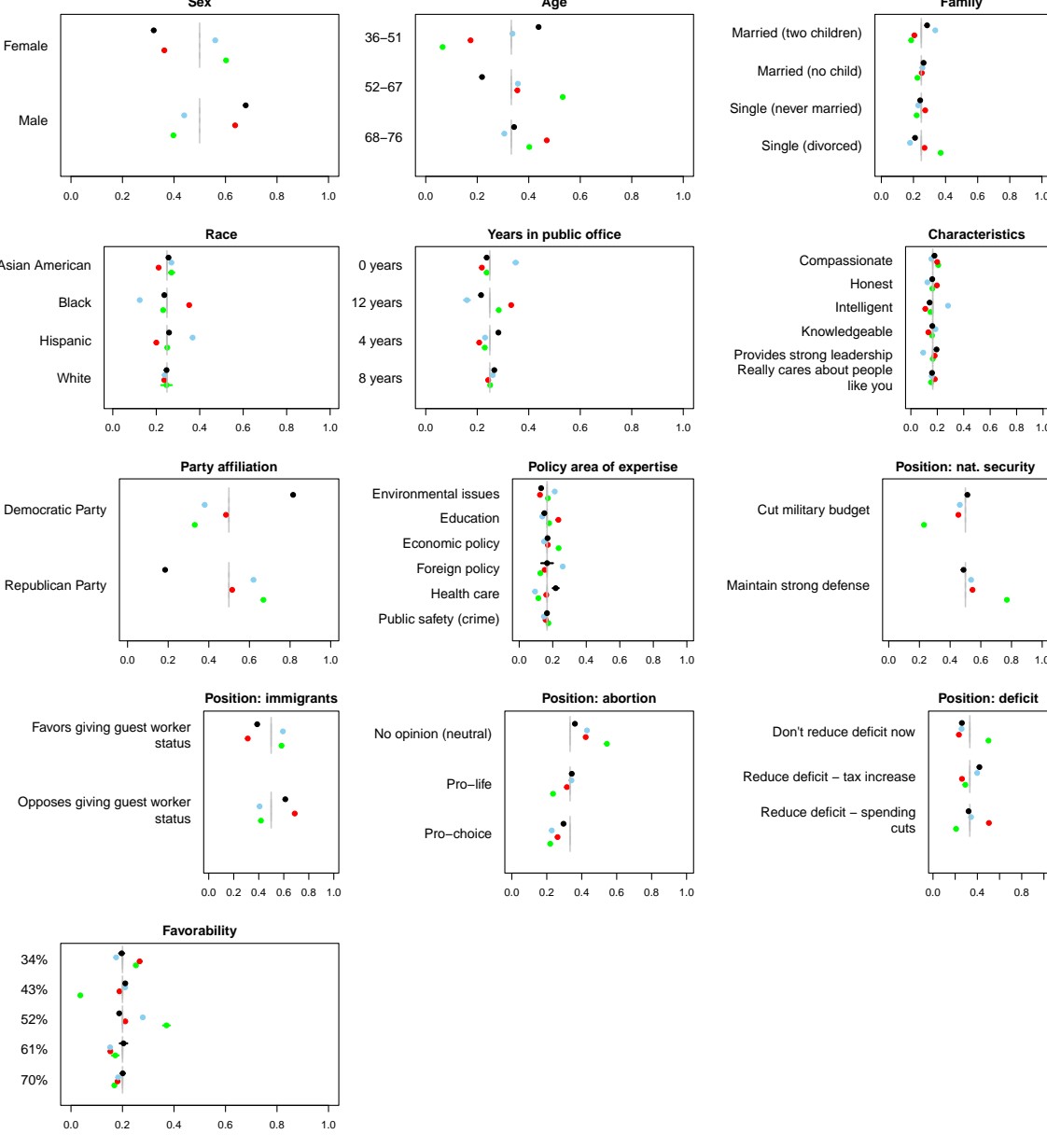

*Figure 13.* Average-case optimal stochastic interventions under the neural outcome model. Panels show the optimal policy among all (black), Democrat (blue), Republican (red), and Independent (green) respondents.

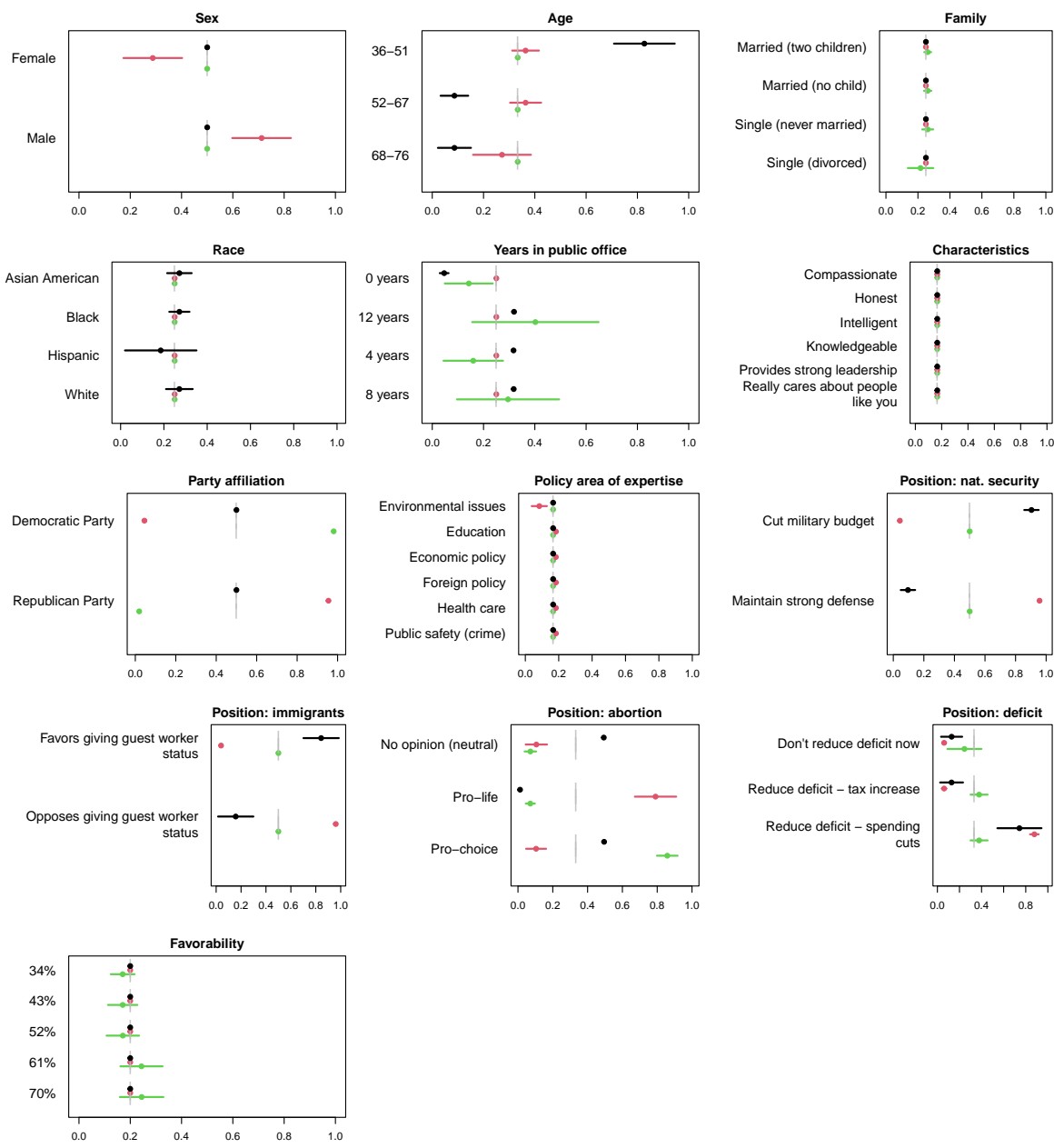

*Figure 12.* Optimal strategies in the covariate sensitive case, where a different strategy for allocating candidate features can be used for three data-derived clusters of voters.

## C.3. Application Results - Outcome Model (GLM)

This subsection reports the U.S. presidential application figures using the GLM-based outcome model approach.

*Table 5.* Mean log-probabilities of 2016 primary candidates under estimated optimal stochastic interventions using the GLM outcome model.

| Party | Quantity | Mean Log Prob. (s.e.) |
|---|---|---|
| Democrats | Average case | -16.05 (0.58) |
| Democrats | Adversarial case | -17.84 (0.49) |
| Democrats | Log likelihood ratio | -1.79 |
| Republicans | Average case | -15.78 (0.33) |
| Republicans | Adversarial case | -17.14 (0.39) |
| Republicans | Log likelihood ratio | -1.36 |

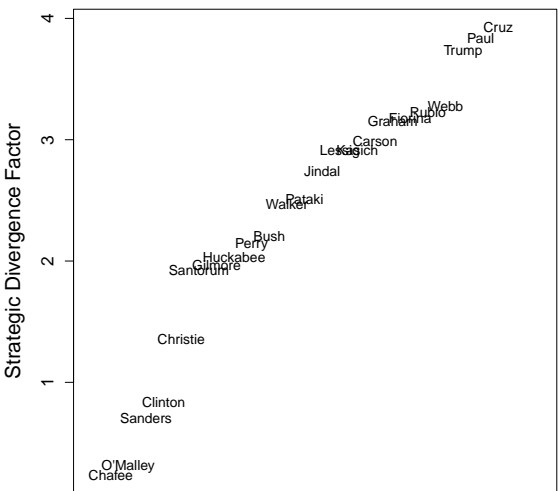

*Figure 14.* Strategic divergence factor computed for major candidates in the 2016 primaries under the GLM outcome model.

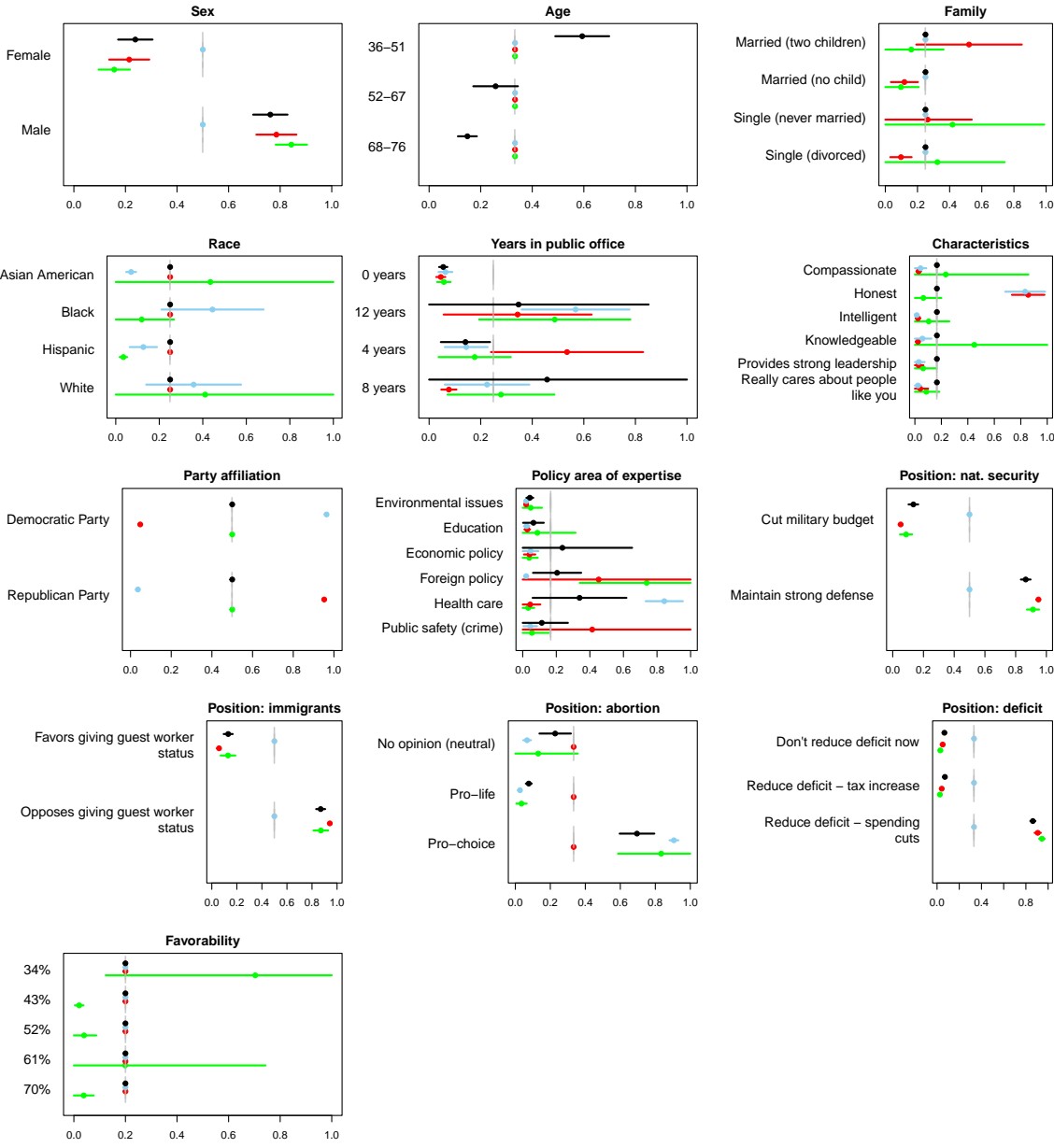

*Figure 15.* Average-case optimal stochastic interventions under the GLM outcome model. Panels show the optimal policy among all (black), Democrat (blue), Republican (red), and Independent (green) respondents.

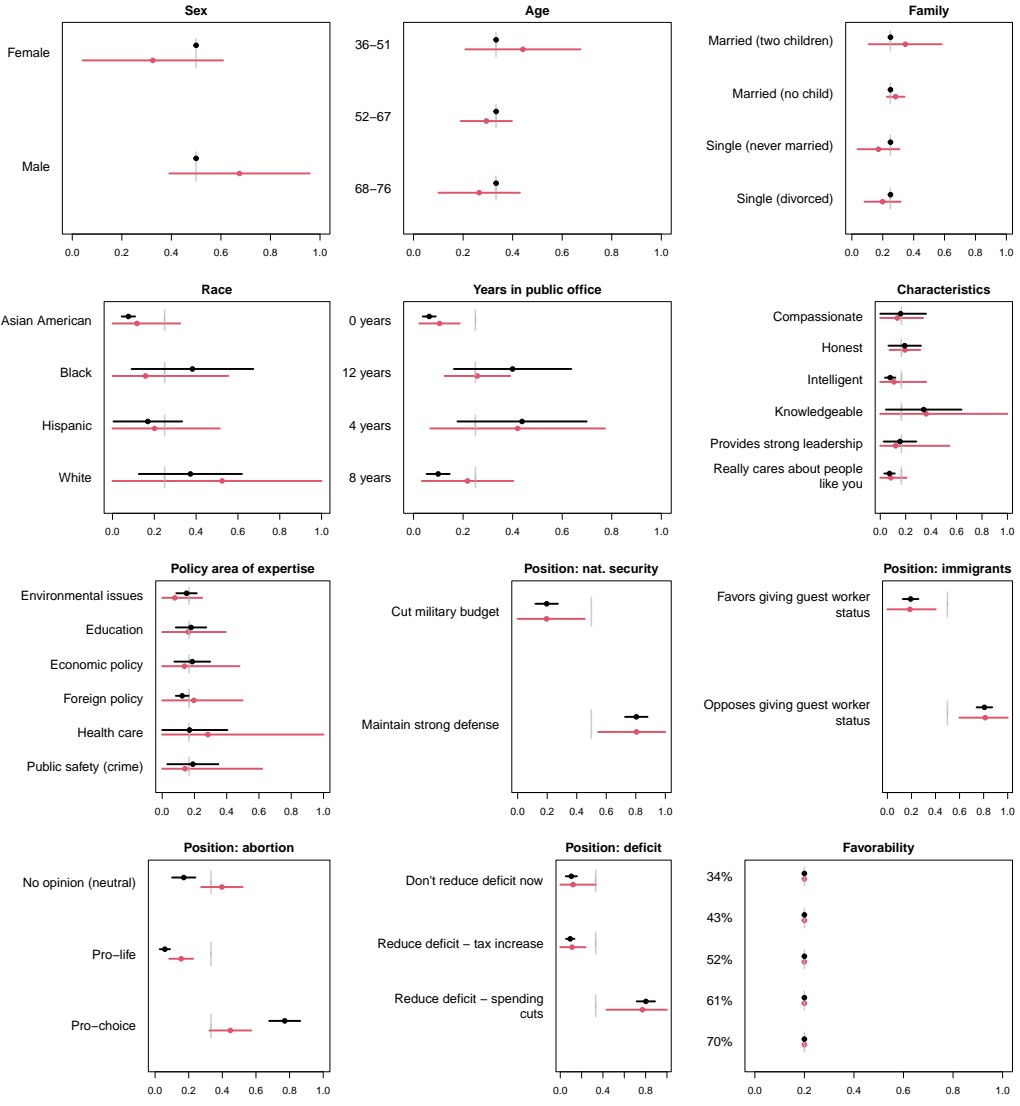

*Figure 16.* Restricted-equilibrium (institution-aware) optimal stochastic interventions under the GLM outcome model.

## C.4. Bayesian Neural Outcome Model

When the linear-in-indicators GLM is too restrictive, a natural modeling choice would be a Bayesian neural approximation in which the pairwise logit is a learned function of the two profiles. In this approach, each candidate profile $\mathbf{t}$ is embedded by a shared Transformer-style encoder over factor-level embeddings (see Fig. 17), yielding a representation $\phi_\theta(\mathbf{t}) \in \mathbb{R}^m$ and a scalar utility $u_\theta(\mathbf{t})$. For a forced-choice pair $(\mathbf{t}^a, \mathbf{t}^b)$, we model the logit as

$$\eta_\theta(\mathbf{t}^a, \mathbf{t}^b) = u_\theta(\mathbf{t}^a) - u_\theta(\mathbf{t}^b)$$
$$+ \omega\{b_\theta(\mathbf{t}^a, \mathbf{t}^b) - b_\theta(\mathbf{t}^b, \mathbf{t}^a)\},$$
$$b_\theta(\mathbf{t}, \mathbf{u}) = \phi_\theta(\mathbf{t})^\top M_\times \phi_\theta(\mathbf{u}),$$

where $M$ is a learned interaction matrix and $\omega$ is a learned scaling factor. This antisymmetrized bilinear term preserves the forced-choice restriction $\eta_\theta(\mathbf{t}^a, \mathbf{t}^b) = -\eta_\theta(\mathbf{t}^b, \mathbf{t}^a)$, and hence $\Pr\{a \succ b\} + \Pr\{b \succ a\} = 1$, while still capturing opponent-dependent comparisons without a full cross-encoder. Identical paired profiles have logit zero and choice probability $1/2$. We place priors on $\theta$ and fit a posterior (e.g., by MCMC or variational inference), so uncertainty for downstream estimands can be propagated either by posterior sampling or by a local Gaussian Delta-method approximation based on the

posterior mean and covariance of $\theta$.

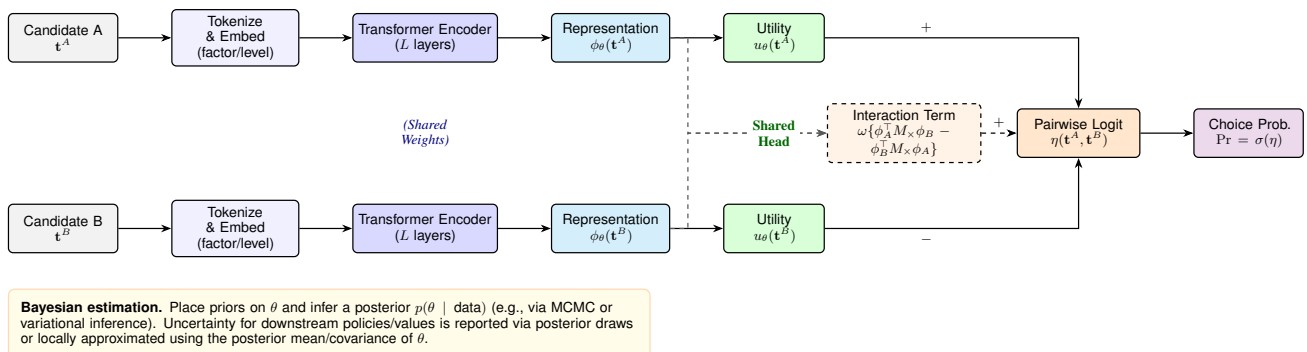

*Figure 17.* Neural outcome model for forced-choice conjoints. Profiles are processed by shared Transformer encoders (blue) to produce latent representations (cyan) and utilities (green). The pairwise logit aggregates utilities and an optional interaction term (orange), mapped to a win probability (purple). All shared weights are indicated by horizontal labels. In our implementation, we use an embedding dimensionality of 64, a depth of 8, and a pre-activation RMS-normalization design.

**Reproducibility.** All methods are implemented in the open-source package `strategize`, available at `https://github.com/cjerzak/strategize-software`. The package uses a `JAX` backend for automatic differentiation and gradient-based optimization, enabling efficient computation on both CPU and GPU. Key hyperparameters include the regularization parameter $\lambda$ (held fixed in experiments for consistency of analyses but in practice selected via cross-validation), the number of ascent/descent iterations (10000), and the number of Monte Carlo samples for value computation (default: 64).

