# OpenReview forum: "MiniMax Learning of Interpretable Factored Stochastic Policies from Conjoint Data, with Uncertainty Quantification"
_ICML.cc/2026/Conference — ICML 2026 regular_

### Official Review · Reviewer_hyNZ · 2026-03-04

**Soundness:** 3
**Presentation:** 3
**Significance:** 3
**Originality:** 3
**Overall Recommendation:** 4
**Confidence:** 2

**Summary:**

This paper proposes a new framework for learning optimal stochastic policies from conjoint survey experiments. Traditional conjoint analysis focuses on estimating marginal effects such as AMCE. In contrast, the authors reinterpret conjoint experiments as an offline policy learning problem over combinatorial action spaces. They introduce a factored stochastic policy class that assigns independent categorical distributions over attribute levels, allowing interpretable distributions over candidate profiles.

The paper develops both average-case and adversarial formulations. In the average-case setting, the objective is to maximize expected outcomes against a fixed reference distribution over opponent profiles. In the adversarial setting, the problem is framed as a two-player zero-sum minimax game, where both parties simultaneously choose distributions over candidate profiles.  The paper provides (i) a closed-form solution for the optimal stochastic intervention under a linear probability model with two-way interactions and L2 regularization, (ii) a general gradient-based procedure for richer outcome models (including neural networks), and (iii) uncertainty quantification via the Delta method. Empirical evaluation includes Monte Carlo simulations, adversarial simulations of two-party competition, and an application to U.S. presidential conjoint data.

**Compliance With Llm Reviewing Policy:**

Affirmed.

**Final Justification:**

I have no further concerns and will maintain the positive score.

**Key Questions For Authors:**

Please see the above parts.

**Limitations:**

yes

**Strengths And Weaknesses:**

Strengths:
1. A key contribution of the paper is the conceptual reframing of conjoint analysis. Instead of estimating marginal causal effects (e.g., AMCE), the authors propose learning optimal stochastic policies over full candidate profiles. This perspective is compelling and may significantly broaden the scope of conjoint experiments.
2. The work effectively integrates ideas from several areas, including causal inference and experimental design, offline reinforcement learning / contextual bandits, game theory and minimax optimization, and political science applications of conjoint analysis. The adversarial extension that models strategic interaction between parties is a natural and interesting extension of traditional conjoint analysis.
3. The paper provides a relatively complete methodological treatment. The work develops a full pipeline including: formal problem formulation, theoretical analysis including closed-form solutions under specific model assumptions, statistical inference for the learned policy via Delta-method uncertainty propagation, scalable optimization procedures for more general outcome models, and extensive empirical validation through both simulation studies and real-world data.

Weaknesses:
Overall, I did not identify serious weaknesses in the paper. Most of the potential limitations stem from standard modeling assumption. For example, the policy class assumes a factored structure where attribute choices are independent, which simplifies the optimization problem and improves interpretability but may restrict the space of possible strategies.  The paper would benefit from providing more discussion or justification regarding these modeling choices and their potential implications in practice.

---

> ### Author Rebuttal · Authors · 2026-03-31
>
> Thank you for the thoughtful review. We appreciate the suggestion to better justify the factorized policy class and to discuss its implications more explicitly.
>
> > "The policy class assumes a factored structure where attribute choices are independent, which simplifies the optimization problem and improves interpretability but may restrict the space of possible strategies."
>
> We agree that this factorized policy class should be justified more explicitly, and that limitations should be stated more clearly: when the true optimal strategy depends on tightly coupled attribute bundles, the resulting policy should be interpreted as the best restricted approximation, and we will make this scope condition more explicit in the revision. (The method is not claiming optimality over all mixed strategies, but rather over an interpretable restricted class, where that interpretability can benefit downstream decision-making or practical deployment.)
>
> While a fully flexible policy class could theoretically capture complex dependencies, estimating such a high-dimensional joint distribution could inflate the variance of the learned policy given outcome model uncertainty. Thus, there may be a tradeoff wherein a more expressive policy class may reduce approximation bias, but at the cost of higher variance, whereas the factorized class may incur some policy-class bias, but yielding more stable uncertainty quantification, especially in the adversarial regime.
>
> To quantify that tradeoff, we are adding a simulation in which the optimal policy distribution will exhibit cross-attribute dependence (using an autoregressive discrete policy network); we will report the resulting policy bias and estimated variance change relative to the restricted factorized policy class.
>
> Again, we appreciate your feedback and think that surfacing the approximation tradeoffs more explicitly will strengthen the paper.

---

> > ### Author Rebuttal · Reviewer_hyNZ · 2026-04-04
> >
> > I have no further concerns and will maintain the positive score.

---

### Official Review · Reviewer_zFKU · 2026-03-11

**Soundness:** 3
**Presentation:** 3
**Significance:** 3
**Originality:** 3
**Overall Recommendation:** 5
**Confidence:** 3

**Summary:**

In previous works, there has been a heavy focus on the Average Marginal Component Effect, which is an estimate of the average causal effect of each feature on preferences while marginalizing the others over some distribution. In this paper, the authors evolve from this framework to a factored stochastic policy learning problem. They contribute a closed-form average-case optimizer for two-way interactions that allows uncertainty quantification for the best policy. They extend this to a minimax setting, and perform simulations on a common application: evaluation of candidate profiles.

**Compliance With Llm Reviewing Policy:**

Affirmed.

**Final Justification:**

Authors have addressed most of my concerns; I am happy to maintain my positive score.

**Key Questions For Authors:**

1.	Figure 2 references a shaded band which I cannot see. As stated before, while the important ideas are clear from the figures, adding color or different line types could be helpful.
2.	Equation 2 is made smaller so that it doesn’t run too long. Perhaps this could be applied to the other equations as well.
3.	How realistic is the linear probability approximation in this setting? The theoretical results rely on it, so a reference to previous literature that has used similar assumptions would suffice.

**Limitations:**

The paper includes a “Limitations” section instead of an “Impact statement.” The authors are very upfront about potential limitations of their method in this section. However, because this paper primarily deals with a presidential election setting, I think it would be worthwhile to discuss potential exploitations of this method or similar methods.

**Strengths And Weaknesses:**

Previous works have rarely looked at cases when factors may have antagonistic interaction, which could be catastrophic. This paper avoids this problem and additionally offers uncertainty-aware confidence intervals for the optimal policy.

Soundness:
This paper offers one main theoretical result in the main manuscript (Proposition 3.1) and provides a proof in the Appendix. Experimental results are also provided for the exact application cited in the introduction using real-world data. The figures presented show the excellent performance of the method in their figures, and the method can achieve a low RMSE with a sample size of 10000. The coverage probability is stable at 95% for this sample size as well. They also provide an example of how the adversarial-case expected outcome appears more realistic than the average-case optimizer.

Presentation:
The paper does a great job of making it clear why this advancement is important. Notation is clear. At some points, the equations run a little long and nearly off the page. The figures are quite small and colorless, which makes it slightly hard to read.

Significance:
AMCE-based policies are naive and can ignore interactions, which can lead to errors as clearly stated in their example and as shown in their experiments. They also offer uncertainty quantification and show that the coverage probability matches the confidence level empirically. This is useful for any preference problem, but the application this paper focuses on is already quite important by itself.

---

> ### Author Rebuttal · Authors · 2026-03-31
>
> We appreciate the constructive review and are glad the contribution and empirical findings came through clearly. We will address the remaining points as follows.
>
> **Figure readability**
>
> > "Figure 2 references a shaded band which I cannot see... adding color or different line types could be helpful."
>
> Thank you; the current rendering is too subtle. In the revision, we will redraw Fig. 2 (the historical vote-share comparison) so the historical range is clearly visible in both color and grayscale (larger text, thicker lines, etc.). We will similarly improve other figures throughout.
>
> **Equation formatting**
>
> > "Equation 2 is made smaller so that it doesn’t run too long. Perhaps this could be applied to the other equations as well."
>
> Agreed. We'll reformat.
>
> **Linear probability approximation**
>
> > "How realistic is the linear probability approximation in this setting?"
>
> This is an important point. The linear-probability approximation is used for the closed-form average-case result; it makes an analytical closed form tractable and clearly exposes the role of the interaction terms. The broader framework is not limited to that assumption: in the paper, we also give gradient-based optimization for Bernoulli GLMs, and in the Appendix we show performance with a Bayesian neural Transformer-style outcome model (architecture outlined in Figure 17). In the revision, we will make this distinction clearer, better highlight these Appendix results, and emphasize references to prior conjoint work where linear models for randomized forced-choice outcomes are standard (e.g., Hainmueller et al., 2014; Egami and Imai, 2019).
>
> To address robustness further, we are also adding a new model-misspecification analysis in which the true choice model includes non-linear structure while estimation still uses the linear + two-way interaction specification or a Bayesian Transformer. We report the effect on policy recovery and interval coverage in Table RR-1, below. These results clarify the steepness of the performance drop under misspecification and the relative robustness of Transformer outcome models in that context.
>
> **Potential misuse / broader impacts**
>
> > "Because this paper primarily deals with a presidential election setting, I think it would be worthwhile to discuss potential exploitations of this method or similar methods."
>
> Misuse is a serious concern that we will expand on, explaining how methods like ours could in principle be used for strategic persuasion or manipulative candidate positioning. We will also explain how some of the diagnostics introduced in the paper (e.g., Eq. 4) could also be used to detect whether such activity is occurring in practice.
>
> ---
>
> ### Table RR-1
>
> In a new model misspecification analysis, we start with a linear outcome surface built from main effects and pairwise interactions. To introduce nonlinearity, we construct nonlinear transforms of grouped treatment averages (sin, cos, log1p, exp), combine them into one latent nonlinear score, then project that score onto the original linear design. We then extract the residual as the truly nonlinear part of the score. We then generate the outcome using the linear and nonlinear components; a scaling factor is chosen using grid search so the linear model’s unexplained variance share (misspec = 1 - $R^2$) matches the target. Outcomes are generated from the linear signal plus that scaled nonlinearity. In this way, the GLM estimation model is faced with misspecification.
>
> We find that, perhaps unsurprisingly, as the degree of misspecification grows, Bayesian transformer performance becomes relatively more favorable compared to the GLM with interactions. However, the neural approach uses a variational approximation, so its uncertainty estimates are less accurate (resulting in reduced coverage). Also, when the degree of misspecification is low, the GLM approach is much more data-efficient than the Transformer (hence, a lower overall RMSE in that case).
>
> ---
>
> _Preliminary results from model misspecification analyses (averaging over sample size $n$ [from 500 to 10000] and number of factors [from 5 to 20]; final disaggregated results to appear in Appendix)._
>
> _GLM outcome model with pairwise interactions:_
>
> | Misspec. | Coverage (Q) | Mean RMSE (Q) |
> |---:|---:|---:|
> | 0.0 | 0.94 | 0.007 |
> | 0.1 | 0.66 | 0.158 |
> | 0.5 | 0.78 | 0.553 |
>
> _Bayesian Transformer outcome model:_
>
> | Misspec. | Coverage (Q) | Mean RMSE (Q) |
> |---:|---:|---:|
> | 0.0 | 0.43 | 0.311 |
> | 0.1 | 0.56 | 0.216 |
> | 0.5 | 0.50 | 0.424 |

---

> > ### Author Rebuttal · Reviewer_zFKU · 2026-04-02
> >
> > Thank you for the response. I will maintain my positive score.

---

> > > ### Author Response · Authors · 2026-04-04
> > >
> > > We sincerely thank the reviewer for the time taken to access our work and are happy to answer any further questions, should they come up.

---

### Official Review · Reviewer_Wkqo · 2026-03-11

**Soundness:** 3
**Presentation:** 2
**Significance:** 2
**Originality:** 2
**Overall Recommendation:** 4
**Confidence:** 2

**Summary:**

The paper proposes a framework for learning interpretable stochastic policies from conjoint experiments by optimizing distributions over high-dimensional candidate profiles instead of estimating marginal feature effects like AMCEs. Using an outcome model estimated from conjoint data, the method learns a factored policy over attributes that maximizes expected outcomes while controlling variance, and propagates uncertainty to the learned policy using the Delta method. The framework is further extended to an adversarial minimax setting where competing agents strategically choose profile distributions. Experiments on synthetic data and a U.S. presidential conjoint dataset show that adversarially optimized strategies produce more realistic electoral outcomes.

**Compliance With Llm Reviewing Policy:**

Affirmed.

**Final Justification:**

I thank the reviewer for the detailed response. I agree that the paper may offer meaningful contributions on its own and derived substational theories. However, it remains unclear to me to what extent these contributions are driven by ML. For example, it appears to me that the cited literature on conjoint analysis appears to be primarily rooted in political analysis rather than the ML domain. I am choosing to maintain my original score.

Update: I appreciate the follow-up response from the authors. Upon a brief look, the cited paper does seem to demonstrate the relevance of conjoint analysis with modern ML, so I am raising my score.

**Key Questions For Authors:**

See weaknesses.

**Limitations:**

Yes

**Strengths And Weaknesses:**

While the paper may have merits, I am unable to fully assess it because I am not familiar with the scope and method of this paper (i.e., conjoint analysis). I am scoring my review with low confidence.

Meanwhile, it appears to me that conjoint analysis seems to be primarily relevant to a narrow subset of the statistics community. I believe the paper may benefit more from the review process of a statistics journal rather than ICML, where experts in this area could provide more specialized and in-depth feedback.

# Strengths:

- The paper has a good and fluent presentation up to the first column of page 3, where I was able to follow most of the contents.

- The paper contributes to making a conceptual shift from marginal causal effect estimation (AMCE) to policy learning over candidate profiles. Thus, the paper reframes conjoint analysis as decision-making under competition, which is intellectually appealing.

- There are many useful components of the proposed methods. For example, the proposed estimator is efficient to compute since it has a closed-form solution. The solution comes with uncertainty quantification by using the Delta method.

# Weaknesses:

- The scope of the paper appears to be more aligned with a specialized subset of the statistics community rather than the broader machine learning audience. It may misalign with the scope of ICML.

- The methodology relies heavily on parametric assumptions. For example, the categorical distribution and Bernoulli GLM. It is unclear whether or not the proposed method would still hold if these assumptions are violated.

- The paper is missing a related work section to summarize existing studies and developments. This makes it difficult to evaluate the novelty and contribution of the work. Also, I suggest that the background section on conjoint analysis be extended with more technical details to help the audience that is unfamiliar with conjoint analysis understand the novelty of the method.

# Other Comments

- Line 262: Proposition 1 is not defined. I think the authors mean Proposition 3.1?

- In the abstract, the full name of AMCE should be defined before presenting its abbreviation.

---

> ### Author Rebuttal · Authors · 2026-03-31
>
> We thank the reviewer for the helpful comments and for highlighting where the paper needs to be clearer for a broader ML audience.
>
> **Scope**
>
> > "The scope of the paper appears to be more aligned with a specialized subset of the statistics community rather than the broader machine learning audience."
>
> We appreciate this concern. We see the paper as relevant to these areas mentioned in the ICML 2026 call:
> ```
> - optimization (convex and non-convex optimization, ..., etc.)
> - trustworthy machine learning (..., causality, ..., interpretability, ..., etc.)
> ```
> But we agree that fit should be judged by the substance of the contribution, not by the application domain alone.
>
> Methodologically, the paper contributes optimization results through a new closed-form solution in a tractable regime and general gradient-based optimization for richer model classes, including adversarial settings. It also contributes to trustworthy ML through interpretable policy learning, causal framing, and uncertainty quantification for learned stochastic (max-min) policies. The conjoint domain is the setting in which the method is motivated and evaluated, but the underlying contribution is arguably more general.
>
> **Parametric assumptions/misspecification**
>
> > "The methodology relies heavily on parametric assumptions. For example, the categorical distribution and Bernoulli GLM. It is unclear whether or not the proposed method would still hold if these assumptions are violated."
>
> The factored product-of-Categoricals policy class is a deliberate choice for interpretability and tractability. The Bernoulli GLM and the linearized objective are used because they speed up estimation and, in one regime, enable a closed-form optimum. But the broader framework is not tied to a single parametric specification: the paper gives a general differentiable recipe for richer GLMs and Bayesian neural outcome models (see Fig. 17 in the Appendix for the architectural design and associated results).
>
> That said, to explicitly quantify the empirical sensitivity of learned policies/values to model misspecification, we are running a new analysis (see Table RR-1 below). There, the true outcome includes nonlinearities while estimation still uses the linear + pairwise-interaction model or the Bayesian transformer architecture.
>
> Separately, we are adding a sensitivity analysis to isolate the approximation-estimation tradeoff induced by the factored policy restriction. We will here compare the interpretable factored product-of-Categoricals class (viewed in Appendix A.6 as a structured approximation to the unrestricted optimum) with a more flexible policy model based on an autoregressive discrete policy network. The latter can reduce approximation bias by capturing cross-attribute dependence, but we expect that added flexibility to increase estimation uncertainty.
>
> **Related work**
>
> > "The paper is missing a related work section... [and] the background section on conjoint analysis [should] be extended with more technical details."
>
> Agreed. We did have a related-work section buried in Appendix B, but it should be moved into the main paper so we can more clearly cover related work. See Author Response to X7Qk for an overview.
>
> **Minor corrections**
>
> Thank you. We will fix.
>
> ## Table RR-1
>
> In a new model misspecification analysis, we start with a linear outcome surface built from main effects + pairwise interactions. To introduce nonlinearity, we construct nonlinear transforms of grouped treatment averages (sin, cos, etc.), combine them into one latent nonlinear score, then project that score onto the original linear outcome. We then extract the residual as the truly nonlinear part of the score. We then generate the outcome using the linear and nonlinear components; a scaling factor is chosen so the linear model’s unexplained variance share (misspec = 1 - $R^2$) matches the target. Outcomes are generated from the linear signal plus that scaled nonlinearity. In this way, the GLM estimation model faces misspecification.
>
> We find that, as the degree of misspecification grows, Bayesian transformer performance becomes relatively more favorable compared to the GLM with interactions. However, the neural approach uses a variational approximation, so its uncertainty estimates are less accurate (reduced coverage). Also, when the degree of misspecification is low, the GLM approach is much more data-efficient than the Transformer (lower RMSE).
>
> ---
>
> _Preliminary results from model misspecification analyses (averaging over sample size $n$ [from 500-10000] and number of factors [from 5-20]; disaggregated results to appear in Appendix)._
>
> _GLM outcome model with pairwise interactions:_
>
> | Misspec. | Coverage (Q) | Mean RMSE (Q) |
> |---:|---:|---:|
> | 0.0 | 0.94 | 0.007 |
> | 0.1 | 0.66 | 0.158 |
> | 0.5 | 0.78 | 0.553 |
>
> _Bayesian Transformer:_
>
> | Misspec. | Coverage (Q) | Mean RMSE (Q) |
> |---:|---:|---:|
> | 0.0 | 0.43 | 0.311 |
> | 0.1 | 0.56 | 0.216 |
> | 0.5 | 0.50 | 0.424 |

---

> > ### Author Rebuttal · Reviewer_Wkqo · 2026-04-03
> >
> > I thank the reviewer for the detailed response. I agree that the paper may offer meaningful contributions on its own and derived substational theories. However, it remains unclear to me to what extent these contributions are driven by ML. For example, it appears to me that the cited literature on conjoint analysis appears to be primarily rooted in political analysis rather than the ML domain. I am choosing to maintain my original score.

---

> > > ### Author Response · Authors · 2026-04-03
> > >
> > > We sincerely thank the reviewer for their time spent assessing our work.
> > >
> > > We realize that our specific empirical focus may have inadvertently given the impression that conjoint analysis is restricted to a single domain. We would like to clarify that the method is widely used in fields where multi-dimensional preferences are important [1]. The method is also closely related to combinatorial learning which appears prominently in RL [2].
> > >
> > > Furthermore, conjoint analysis has a precedent across various machine learning domains. Some examples:
> > >
> > > Wang et al. (ICML 2024) introduces methods for quantifying total variation of conjoint features on outcomes via Total Variation Floodgate.
> > >
> > > Chapelle & Harchaoui (NeurIPS 2004) develops a machine learning approach to conjoint analysis.
> > >
> > > Nie et al. ( NeurIPS 2023) deploys conjoint analysis to study LLM alignment.
> > >
> > > Dudik et al (AISTATS 2012) develops ML methods for conjoint analysis from a multi-task learning perspective.
> > >
> > > Finally, the core contributions of our work are methodological rather than domain-specific: offline policy learning over combinatorial action spaces, closed-form and gradient-based optimization for learned stochastic policies in average-case and minimax settings, and uncertainty quantification for those learned policies. Conjoint data are the experimental substrate in our application, but the estimand, optimization, and inference machinery are we think more general.
> > >
> > > ## References
> > > [1] Agarwal et al. An interdisciplinary review of research in conjoint analysis: Recent developments and directions for future research. Customer Needs and Solutions. 2015.
> > > [2] Lee & Oh. "Combinatorial Reinforcement Learning with Preference Feedback." ICML 2025.

---

### Official Review · Reviewer_X7Qk · 2026-03-13

**Soundness:** 3
**Presentation:** 2
**Significance:** 3
**Originality:** 3
**Overall Recommendation:** 5
**Confidence:** 2

**Summary:**

This paper frames optimal profile selections as a factored stochastic policy learning problem, shifting from the traditional AMCEs methods. This setup captures the strategic interdependence and reserves attribute-level interpretability. The authors provided a closed-form solution average-case solution as well as a gradient-based algorithm for general setups. The authors also considered the adversarial settings and provided a minimax extension. The proposed framework and methods are verified by various numerical experiments.

**Compliance With Llm Reviewing Policy:**

Affirmed.

**Final Justification:**

All concerns are addressed. Therefore, I keep my positive score.

**Key Questions For Authors:**

**1.** How does the theoretical guarantee in this paper compare with previous works in similar settings?

**Limitations:**

The limitations in this work are the same as listed in "Key Questions For Authors", addressing or clarifying these questions could strengthen this paper significantly.

**Strengths And Weaknesses:**

**Soundness**
*Strengths:*  The claims in this paper are well supported by closed-form solutions or asymptotic behavior guarantees. The effectiveness of the proposed methods are also verified by various experiments.

**Presentation**
*Strengths:* The paper is generally clear and well structured, the positioning and comparisons to previous works are clear.
*Weaknesses:*
Each section (especially Methods) could use several sub-sections for clarity.
Also, the formatting of the equations should be carefully revised.

**Significance and Originality**
This paper proposed the framework to use factored stochastic policy learning for optimal profile selections, which is a significant improvement compared to traditional methods such as AMCE. The authors also proposed methods to estimate uncertainty and strategies for the adversarial setting. Overall, this paper is novel and significantly contributes to this field.

---

> ### Author Rebuttal · Authors · 2026-03-31
>
> Thank you for the comments and supportive assessment of the paper. We appreciate the request to better situate the theory and to improve the presentation.
>
> **Comparison to prior theoretical guarantees**
>
> > "How does the theoretical guarantee in this paper compare with previous works in similar settings?"
>
> We will clarify this comparison more explicitly in the main text (pulling content from a related work subsection in the Appendix). The closest bodies of literature provide guarantees that differ in that they primarily focus on marginal causal quantities in conjoint designs, policy learning in lower-dimensional regimes, or max-min optimization with known utilities; our work connects them:
>
> 1. **Conjoint methodology (AMCE/AMIE).** Prior theory primarily characterizes marginal causal effects under randomized conjoint designs [1,2]. Those results do not to our knowledge characterize an interpretable optimizer over profile distributions, nor uncertainty for the learned optimal policy itself. Our paper instead studies this policy object -- a factored stochastic distribution over full profiles -- and derives uncertainty for both the learned policy $\pi^{\ast}$ and its value $Q(\pi^{\ast})$.
>
> 2. **Offline policy learning/treatment rules.** Much of the classical econometric policy-learning / treatment-rule literature studies binary or otherwise small action spaces [3,4], with later continuous-treatment [5] and (pure RL) multi-action extensions [6]. Our setting has an exponentially large combinatorial action space. In that structured regime, we exploit the factorization to obtain a closed-form optimizer under two-way interactions + L2 regularization, along with Delta-method asymptotics for both the optimal policy and its value (valid for generalizations beyond two-way interactions). So the contribution is more specific than general policy-learning theory, but stronger for this high-dimensional conjoint setting and in modeling in adversarial behavior.
>
> 3. **Minimax/zero-sum learning.** Classical results give existence [7] and convergence [8] guarantees on the full mixed-strategy simplex. Our contribution is to adapt this perspective to interpretable factored policies aligned with conjoint designs, and to propagate first-stage outcome-model uncertainty into the learned restricted minimax solution. This, to our knowledge, has not been studied previously. We will make it clearer in the revision that, in the adversarial setting, our guarantees are for the restricted/differentiable optimization problem rather than a universal claim over the full simplex.
>
> Beyond moving the Related Work section to the main text, we will also add a concise contributions paragraph in the main text that distinguishes: (i) exact closed-form optimality in the tractable two-way regime, (ii) asymptotic normality for $\pi^{\ast}$, $Q(\pi^{\ast})$ under regularity conditions, and (iii) stationary-point / first-order guarantees for the general differentiable optimization case.
>
> **Presentation**
>
> > "Each section (especially Methods) could use several sub-sections for clarity."
>
> Noted. We will reorganize the Methods section into shorter subsections with a more explicit roadmap: setup/estimand, average-case objective, closed-form solution, general gradient-based optimization, adversarial extension, and uncertainty quantification. We will also move some denser derivations or theory statements to the Appendix while adding short intuition paragraphs before the most technical blocks.
>
> > "The formatting of the equations should be carefully revised."
>
> Agreed. We will revisit equation formatting throughout for readability.
>
> ## References
>
> [1] Hainmueller et al. Causal inference in conjoint analysis. Political Analysis, 22(1):1-30, 2014.
>
> [2] Egami & Imai. Causal interaction in factorial experiments. JASA, 114(526):529-540, 2019.
>
> [3] Zhao et al. Estimating individualized treatment rules using outcome
> weighted learning. JASA, 107(499):1106-1118, 2012.
>
> [4] Athey & Wager. Policy learning with observational data. Econometrica, 89(1):133-161, 2021.
>
> [5] Kallus & Zhou, 2018. Policy evaluation and optimization with continuous treatments. AISTATS. 84:1243-1251.
>
> [6] Zimmert & Seldin, 2018. Factored bandits. NeurIPS, 31.
>
> [7] Von Neumann, 1928.
>
> [8] Nemirovski, 2004. Prox-method with rate of convergence O (1/t) for variational inequalities. SIAM Journal on Optimization, 15(1), 229-251.
>
> and others cited in Section B.1 (p. 20).

---

> > ### Author Rebuttal · Reviewer_X7Qk · 2026-04-04
> >
> > All concerns are properly addressed in detail. Therefore, I keep my original score.

---

### Decision · Program_Chairs · 2026-04-30

**Decision:**

Accept (regular)

**Comment:**

This submission studies offline learning of interpretable factored stochastic policies from conjoint data, with both average-case and minimax formulations and accompanying uncertainty quantification. This paper makes a nice methodological contribution. In particular, the paper goes beyond standard marginal conjoint estimands by formalizing policy learning over combinatorial action spaces and deriving tractable optimizers.

The reviews were mixed but tilted positive after rebuttal. Most technical and presentation concerns were addressed in the discussion. The main remaining limitations concern scope and positioning rather than soundness: the paper should use the camera-ready revision to better articulate when the factorized/parametric assumptions are appropriate, sharpen its comparison to adjacent literatures, and make its machine-learning relevance more explicit. With those revisions, I view this as an above-bar ICML paper and recommend Accept.